# Advancing Nearest Neighbor Explanation-by-Example with Critical Classification Regions

## Abstract

There is increasing evidence suggesting post-hoc explanation-by-example with nearest neighbors is a promising solution for eXplainable Artificial Intelligence (XAI). However, despite being researched for decades, such methods have never seriously explored how to enhance these explanations by highlighting specific *parts* in a classification. Here, *Critical Classification Regions* (CCRs) are proposed to do this, and several methods are compared to determine the best approach. CCRs supplement explanation-by-example by highlighting an important *part* in a test image, with where it was "learned" from in the training data. Experiments across multiple domains show that CCRs represent key features used by the CNN in both the testing and training data. Moreover, a suitably-controlled user study (N=163) on ImageNet, shows CCRs improve people's assessment towards the correctness of a CNN's predictions for misclassifications due to ambiguity.

## 1 Introduction

The impressive success of artificial neural networks (ANNs) has led to proposals they should be used in high-stakes applications such as diagnostic radiology for Covid-19 (Pham, 2020). However, interpretability issues for these models raise significant questions about their feasibility for such use-cases. Accordingly, many eXplainable AI (XAI) techniques have been proposed to overcome this, such as saliency maps (Zhou et al., 2016) and contrastive methods (Miller, 2019). Here, the focus is on one of the most long-standing XAI techniques, the use of post-hoc explanation-by-example with nearest-neighbors (Sørmo et al., 2005), with a view to extending its functionality. Recently, explanation-by-example techniques have been increasingly used to interpret deep learning models (Lipton, 2018; Chen et al., 2019; Jeyakumar et al., 2020; Kenny & Keane, 2021), with interest being bolstered by their psychological plausibility in human decision-making (Klein, 1989), human category-learning (Edwards et al., 2019), and supportive evidence from several user studies (Borowski et al., 2020; Kenny et al., 2021; Buçinca et al., 2020). Currently however, these techniques don't connect important "parts" used in a prediction, we rectify this here.

This work has two main aims: (1) to extend explanation-by-example's functionality through the use of *Critical Classification Regions* (CCRs), and (2) to properly test this novel technique in a suitably-controlled user study.[1] CCRs represent the primary region of importance in a test image and in the nearest-neighbor used for explanation (e.g., see Fig. 1). Although many post-hoc XAI techniques for deep learners show salient regions of a test image (Ribeiro et al., 2016; Bach et al., 2015; Zhou et al., 2016), to our knowledge, none have seriously considered relating these regions to nearest neighbors in the training data, to show *what* features were learned by the model and (more importantly, our core novelty here) *where* they arose.

In the XAI literature, explanation-by-example is perhaps the most popular XAI method (Keane & Kenny, 2019), with large user support (Buçinca et al., 2020). However, few works address the identification of important regions within example-based explanations. The closest work to ours makes use of Feature Activation Maps (FAMs) within the *twin-system framework* to enhance example-based explanations (Kenny & Keane, 2021). FAMs upsample the most positively contributing convolutional feature kernel (in the last convolutional layer) and, as such, can be thought of as a variant of Class Activation Maps [CAMs by Zhou et al. (2016)], which use a combination of all layers.

---

[1]Our method is available to pip install as a Python library at https://after_anon_review.

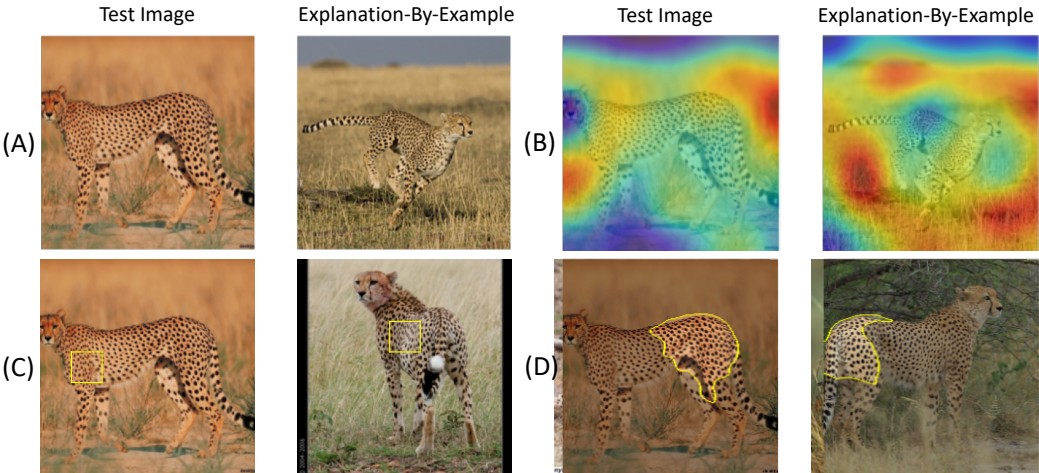

Figure 1: Different Methods: A misclassification of "Leopard" as "Cheetah" on ImageNet is explained by: (A) Explanation-by-example saying *"I think the test image is a Cheetah because it looks similar to a training image I also think is a Cheetah"*. This is then enhanced by (B) FAMs by Kenny & Keane (2019), and our proposed methods of (C) Latent-based CCRs, and (D) pixel-based CCRs. The FAMs highlight multiple regions in both images, and it is difficult to know where a feature begins and ends, in contrast, CCRs highlight a very specific *part* of both images. Note that the CCR explanation methods naturally retrieve a different neighbor due to searching a "pool" of candidates for the closest feature match, whilst FAMs always use the closest NN.

However, FAMs often fail to isolate a comprehensible "part" of each image, and can be hard to contextualize (see Fig. 1B). Moreover, there is no computational tests of FAMs, or user studies. In contrast, we show how CCRs play a key role in classification, and how they affect users in a study.

There is also a literature on post-hoc explanations which isolate prototypical examples from training data (Kim et al., 2014; 2016). These bear a similarity to the present work, but they do not focus on isolating important parts in the classification which we do here. There is also work in the computer vision attention literature such as that by Patro & Namboodiri (2018), which also use exemplars to guide feature highlighting. However, this has similar issues to FAMs, and requires a specific ANN architecture for a very specific task, whilst CCRs are agnostic for ANN vision models. Next, this work bears some similarity to Goyal et al. (2019), but their work focuses on counterfactual explanations, showcasing an algorithm which modifies the smallest part possible to modify a classification, whilst ours is on factual explanations, showing why a classification was made in the first place (the two have very different algorithmic goals). Finally, note also that none of these methods consider superpixels as we do here.

From an implementation standpoint, the current latent-based CCR method (see Section 2.2) is partly inspired by Chen et al. (2019), who used a case-based approach to explain a CNN's predictions. However, their approach is a *pre-hoc* method that compresses the training data down into prototypical parts for explanations, integrating them directly into the CNN's calculations in a forward pass. In contrast, the current method is *post-hoc*, using the training data for explanation *after* the prediction is made. Hence, it may be better suited to situations where (i) model accuracy is critical, (ii) it is simply not feasible to train another more interpretable model, or (iii) more diverse explanations (from the full training data) are required (Mothilal et al., 2020). Finally, the current method builds upon the family of activation-mapping techniques for visualizing the important regions of an image used by a CNN in its predictions (Zhou et al., 2016; Kenny & Keane, 2021).

This paper has four more sections. Section 2 describes how to compute CCRs in explanation-by-example. As CCRs can be generated from the latent space (i.e., Latent-based CCRs), or the pixel space (i.e., Pixel-based CCRs), Section 3 determines which option works best for the *test image*. Section 4 performs additional checks on CCRs to help validate their role in classification, and determines the best way to identifying CCRs in nearest-neighbor explanations (henceforth the

NN-CCR) in the *training image*. Section 5 reports a user study that tests whether the CCRs found by this method matter to people, before a final discussion (see Section 6). This paper makes two main contributions:

- A new XAI idea that computes CCRs to augment *post-hoc*, nearest-neighbor explanation-by-example. These CCRs *connect* critical features used in a classification with *where* they were learned from in the training data (to e.g. help detect bias and aid in explanation). A method for CNNs is proposed (latent-based CCRs), and a "black-box" method (SP-CCRs) which is ANN agnostic for computer vision tasks.
- A novel user study design which is heavily controlled (e.g., with counterbalancing) to specifically test the difference between explanation-by-example, and explanation-by-example with feature highlighting (i.e., the CCRs).

## 2   ALGORITHM: FINDING CCRS IN EXPLANATIONS

Taking an explanation-by-example method, our algorithm begins by finding a pool of $k$ nearest neighbors (we use $k = 50$), all of which will be searched to find the NN-CCR.[2] Two general approaches are considered, one which takes advantage of a typical CNN model's architecture (i.e., the Latent-based CCRs next), and one which makes no assumptions other than the model being a neural network which extracts a learned representation (i.e., Pixel-based CCRs). For the algorithm pseudo-code see Appendix E (omitted due to space constraints). An important points to note about CCR methods is that the regions found are contingent on the pool of NNs over which they are computed (hence, the need for the test of $k$-NN methods reported in Appendix A).

### 2.1   COMPUTING LATENT-BASED CCRS (CAM-CCRS, FAM-CCRS, RAND-CCRS)

This CCR Method takes inspiration from Chen et al. (2019), but crucially, in a *post-hoc* (rather than their *pre-hoc*) manner, specifically, it uses the latent representations of an image in a CNN to gauge similarity. Assume for a given test image $I$, a final representation $C \in \mathbb{R}^{(h,w,d)}$ is extracted after all convolutional layers, were $h$ and $w$ represent the height and width of the convolutional output, respectively, and $d$ the number of kernels (in all current experiments $h = w = 7$). This may be broken down into regions shaped as $h_1 \times w_1 \times d$, were $h_1 < h$ and $w_1 < w$. These regions may be upsampled to the size of the original image to visualize them as a "box" (e.g., see Fig. 1), which is the region in pixel-space that corresponds to this region in $C$.[3] Here, we want to select the region in $C$ which is critical to the classification, and use it to elaborate the explanation further.

Here we set $h_1 = w_1 = 1$ to give more granular detail in the explanation. To select which regions of $C$ are CCRs, we assume the presence of some activation map giving a saliency value to each region in $C$ (e.g., FAMs or CAMs). Such a technique gives a $7 \times 7$ saliency map corresponding to $C$ we refer to as $M_{test} \in \mathbb{R}^{(7,7)}$, which gives the importance of each spatial region. Next, by selecting the most positive salient region, we can isolate the test image CCR $\omega_{test} \in \mathbb{R}^{(1,1,d)}$.

Then, with the test-image CCR isolated, the task is to find a similar region from the training data $\omega_{nn}$, giving an example of where this feature was "learned" from. Using some nearest neighbor algorithm, a pool of $n$ training-instances are found that are candidate explanations-by-example. In this pool, let $C_{(n,i,j)}$ represent some potential region $\omega \in \mathbb{R}^{(1,1,d)}$ in the final convolutional layer, with the neighbor indexed by $n$, and its spatial position in $C_n$ indexed by $i$ and $j$. These $n$ images have their $C$ representation searched to find the closest match to $\omega_{test}$ using the $L_2$ norm to find $\omega_{nn}$. Importantly however, this region in the NNs is constrained to its relative importance within the instance. Specifically, considering each NN's activation map $M_n$, only those regions which satisfy the constraint of being higher than $max(M_n) \times \alpha^{-1}$ are considered to find $\omega_{nn}$ by minimizing:

$$\underset{n,i,j}{\arg\min} \quad \|\omega_{test} - C_{(n,i,j)}\|_2 \qquad \text{s.t.} \quad M_{(n,i,j)} > max(M_n) \times \alpha^{-1}. \tag{1}$$

---

[2]Note we used twin-systems by Kenny & Keane (2021) as it did well in Appendix A tests, and it works for self-supervised learning (which may be more general for future research into CCRs).

[3]Note this could be a bigger box, but we choose a small one to give more granularity. However, more CCRs can be shown if desired (see e.g., Fig. 11), or SP-CCRs used which can give bigger regions.

The alpha constraint is to help ensure $\omega_{nn}$ is highly critical to the classification. For $M$, we experiment using CAM, FAM, and Random maps. So three variants of latent-based CCRs are tested – CAM-CCRs, FAM-CCRs, and Rand-CCRs – although other options exist for retrieving $M$.

## 2.2 Computing Pixel-based CCRs (SP-CCRs)

These CCRs use pixel representations, this is better were spatial relations between the latent layers (i.e., $C$ in the previous section) and the pixel input are not reliable (e.g., because of max-pooling), or convolutions are not used [e.g., vision transformers (Dosovitskiy et al., 2020)], as it is ANN agnostic. Specifically, a test image is broken down into superpixel segments, each segment can be passed into the network one-by-one (with the rest of the image occluded), and the CNN's logit value for the class in question is recorded to find the most important region used in a classification [similar to LIME by Ribeiro et al. (2016)]. Once the test image CCR is isolated, we upsample it to the CNN input-size whilst maintaining its aspect ratio, and acquire its representation from taking the penultimate layer activations after a forward pass in the network. This process is repeated in the pool of NNs, to find the segment (i.e., the NN-CCR) most similar to the test image CCR.

Formally, consider the test image CCR representation $\omega_{test} \in \mathbb{R}^{(d)}$, were $d$ is the number of extracted features in the penultimate layer. We want to isolate a region in the training data $\omega_{nn}$ where this feature was "learned". Let $S_{(n,i)}$ be the representation $\omega \in \mathbb{R}^{(d)}$ of a superpixel segment $i$ in the NN $n$. Additionally, let $M_{(n,i)}$ represent each region's saliency. The $n$ images have their $S$ representations searched to find the closest match to $\omega_{test}$ using the $L_2$ norm to find $\omega_{nn}$. Importantly, this region is constrained to its relative importance within the instance. Specifically, only regions whose saliency is higher than $max(M_n) \times \beta^{-1}$ are considered to find $\omega_{nn}$ by minimizing:

$$\underset{n,i}{\arg\min} \quad \|\omega_{test} - S_{(n,i)}\|_2 \qquad \text{s.t.} \quad M_{(n,i)} > max(M_n) \times \beta^{-1}. \qquad (2)$$

The beta constraint helps ensure $\omega_{nn}$ is highly critical to the classification. For pixel-based CCRs, we consider this single method (i.e., SP-CCRs), but comparative tests against LIME are considered in Expt. 1, but the computational cost of Expt. 2 doesn't allow comparisons to LIME. So, including the three latent-based CCR methods, there are four main methods tested.

## 3 Experiment 1: Finding Test Image CCRs

Here, the performance of the latent- and pixel-based CCR methods in isolating an important part of the *test image* was evaluated. This worked via two methods, by (1) keeping the CCR region in the pixel image (whilst occluding the rest), and (2) by removing the CCR regions (and keeping the rest), followed by passing the image through the CNN in either case. So, in the first case, the method that produces the highest logit in the test-image's predicted class does best, and in the second the method which produces the highest drop in the predicted class logit is best (if the prediction changes we do not change which logit is recorded). In more detail, the test image SP-CCR is located, then the latent-based CCRs are found by up-sampling their activation maps to the pixel-space, and isolating an equally sized region to the SP-CCR in the pixel space taken from the parts of highest saliency (so all CCRs are the same size). Two datasets CUB-200 and ImageNet were used, with the former having ResNet18 fine-tuned to it, and the latter using a pre-trained ResNet50 (see Appendix C). The experiment is repeated with different segmentation options for superpixels (i.e., 10-50) to understand its effect on explanations. Tests used the first 500 test images. Finally, a comparison with LIME is considered to see if it has a significant difference to the SP-CCR algorithm.

## 3.1 Results

Fig. 2(A/B) shows the results of occluding (Occ.) the test image CCR, and Fig. 2(C/D) of including (Inc.) it (and occluding the rest of the image). The top row shows the results of comparing the four CCR methods discussed in Section 2, whilst the bottom row shows comparisons of the latent-based CCR methods against LIME. Note these LIME comparisons correspond to roughly 5% of the test image being isolated as a CCR, which in turn corresponds to roughly 30 superpixels being used in the top row tests. Overall, the results show that CCRs and LIME are significantly better than random. Superpixels perform best for inclusion rather than occlusion when the segment number is

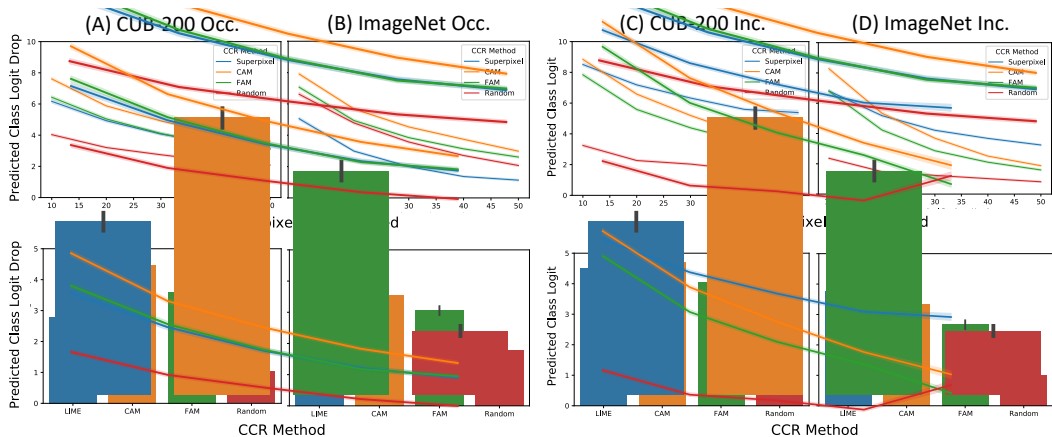

Figure 2: Expt. 1 CCR Occlusion (Occ.) and Inclusion (Inc.) Results: The first row of lineplots show a comparison of the four different CCR methods proposed in Section 2. The second row show a comparison of the three latent-based CCR methods against LIME, to see the difference between LIME and SP-CCRs. Overall, results show the CCR methods do significantly better than random occlusion/inclusion, and that LIME performs similarly to SP-CCRs. Standard Error bars are shown.

greater than 30, but CAMs/FAMs are more consistently good in all tests, although in general FAMs are slightly less discriminatory than CAMs. Perhaps the most notable aspect is how all CCRs (and LIME) do worst when occluding in ImageNet. We posit this is likely because ImageNet has many objects in an image which are used for classification, and removing a small region has little effect. Superpixels in particular do bad here (i.e., worse than random), likely because they still generally maintain the "shape" of the object during occlusion, whilst the latent-based CCR methods (including random) always occlude smoother shapes which distort the objects more. We feel this hypothesis is true because this phenomena was (1) not repeated on CUB-200 (which only has a small part of the image used for classification in comparison), (2) not repeated in the inclusion experiments, and (3) consistent across SP-CCRs/LIME, which is notable because LIME is generally thought to deliver good explanations (Ribeiro et al., 2016; Jeyakumar et al., 2020). So, taking the results as a whole, it is safe to posit that all CCR methods work well in isolating important parts of the test image.

## 4 EXPERIMENT 2: EXPLANATION FIDELITY WITH NN-CCRS

Potential CCRs which are "used" in classification are found by occluding parts of the training data during fine-tuning to see what is necessary to maintain test performance. First, $\alpha$ and $\beta$ are varied from one to infinity [the latter of which is treated as considering all positive CCRs in Eq.(1/2)] to see their optimal value, then a comparative test between the two is done. Eq. (1) and (2) cannot be directly compared because the alpha and beta constraints are relative measures. However, a comparison can be accomplished by (1) varying alpha, (2) closely matching the occlusion area by gradually introducing superpixels (in order of highest saliency value), and (3) readjusting the size of the latent-based methods area to match the superpixel area.

So, for each hyperparamter value tested, the networks were fine-tuned for 2500 iterations and test-accuracy sampled every 50, as this was found sufficient for convergence (we tested 20 epochs and found no notable differences). This procedure gives us an indication of *which regions of the training data images* are actually responsible for test predictions (and what $\alpha$ and $\beta$ are best), whilst the Appendix A experiment gave us an indication of *which training examples* are most appropriate for explanations. The networks were also tested by completely occluding all training images and was reduced to a random guess, thus verifying that features are being "unlearned". Note this experiment requires a constant value for superpixel segmentation, so 30 was chosen as it was the smallest value which generalized best in Expt. 1. Finally, using the best hyperparameters, we can finally test how "similar looking" the NN-CCR is to the test-image CCR by passing the upsampled CCRs though the CNN, and comparing their latent representations in the CNN's penultimate layer with the $L_2$ norm.

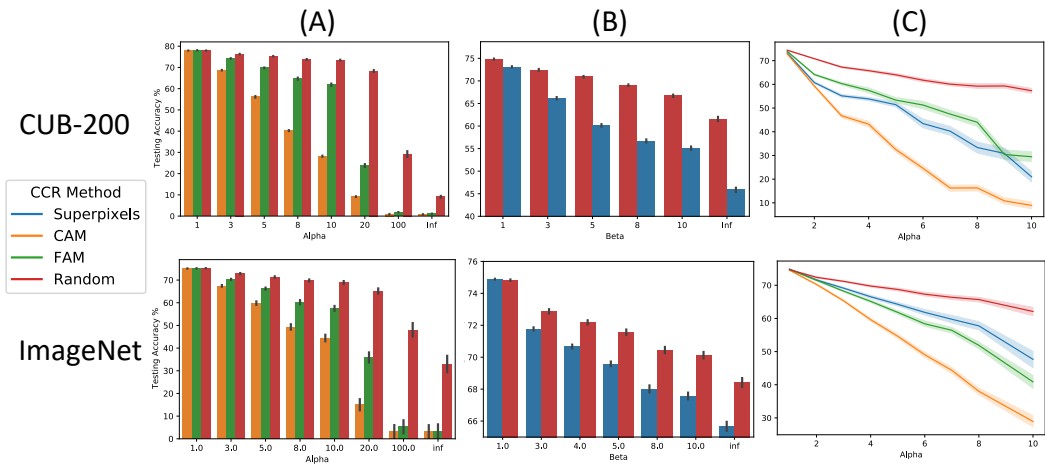

Figure 3: Expt. 2 Results Retraining: (A) Exploring the hyperparameter choice for alpha, (B) doing similar tests for beta. (C) A direct comparison between methods by varying alpha. Results show that all methods are better than the random baseline in nearly every test, and that CAM seems the best method to isolate important critical regions in an image.

### 4.1 RESULTS

Fig. 3(A) shows the results of varying alpha, were any value greater than 1 produces statistically significant differences between methods (using 2 tailed independent t-test; $p < 0.05$). Fig. 3(B) shows what happens to superpixels v. random when varying the beta parameter, were an infinite value (i.e., defined as just using all positive superpixel regions here) produces the most divergent results for CUB-200 (Acc. Rand=61.55 v. SP-CCR=45.86) and ImageNet (Acc. Rand=68.41 v. SP-CCR=65.68). Note that even with $\beta = inf$ only ~66% of the images are occluded in ImageNet on average for SP-CCRs, which roughly equates to the amount occluded for CAM-CCRs at $\alpha = 5$, so there is actually not a huge disparity between them in test accuracy when considering this. For further investigation, Fig. 3(C) shows a direct comparison between methods. Specifically, Fig. 3(C) shows that CAM performs the best overall, with FAMs and superpixels being somewhat interchangeable, but all methods do significantly outperform the random baseline, showing that all method's explanations are likely high in fidelity. Next, each method's distance of the latent representation between the test image CCR and the NN-CCR was compared using $\beta = \inf$ and $\alpha = 5$, showing that the SP-CCRs are significantly better than all other methods when using more than 20 segments (mean SP-CCR $L_2$=14.72/13.4 at segments 20/50 v. CAM mean $L_2$=15.1 on ImageNet). Finally, it should be noted these are empirical tests on retrained networks and should be treated with some caution as they are different to the original CNN being explained, but similar tests have been done before and widely accepted (Hooker et al., 2019).

**Computational Conclusions.** Overall, the results of Expts. 1-2 show that all CCR methods (SP-CCR, CAM-CCR, FAM-CCR) perform better than a random baseline, lending strong empirical evidence they are isolating important regions in classifications. Superpixel segmentation of approximately 30 segments is recommended because it generalizes best across all tests and datasets, forms reasonably large areas which are assumed to aid interpretability, and "look" very similar when comparing the test image CCR to the NN-CCR. For beta an infinite value is recommended, which is computed here as considering all positively contributing SP-CCRs (and not undefined as would be its strict mathematical definition). For alpha, any value greater than one is good, but we recommend five because it worked well here, will tend to focus more on especially important regions, and has support in other research (Zhou et al., 2016). Lastly, it is worth noting that latent-based CCRs are much faster than SP-CCRs (~2sec v. ~45 for $k$=10 neighbors), but SP-CCRs are not restricted to CNNs, which gives them wider applicability (but we test them on CNNs here for a fair comparison). The subsequent user study evaluates CAM-CCRs with $\alpha = 5$.

## 5 EXPERIMENT 3: USER STUDY

The previous computational experiments showed that the current explanation-by-example method with critical regions bears a high fidelity to what the CNN classifier has learned. However, we do not know whether these critical regions matter to people; whether the provision of example-plus-CCR explanations impact a user's perception of the CNN more than example-only explanations. Hence, a user study (N=163) was run to test example-plus-CCR versus example-only explanations, using latent-based CAM-CCRs. Previous user studies have shown that example-based explanations (i.e., based on 3 nearest neighbours to a test-image) changed people's perceptions of the *correctness* of misclassifications by a CNN on MNIST (Kenny et al., 2021). Specifically, they found that people rated misclassifications as less incorrect (i.e., correctness was rated higher). Here, we examine whether the provision of similar explanatory examples with or without CCRs (i.e., a NoBox v Box manipulation) changes people's perceptions of misclassifications made by a CNN on ImageNet.

The XAI literature has few user studies attempting to test *specific* explanation strategies, and most of these are inadequately designed [e.g., poor test-item selection and low Ns are common deficits (Anjomshoae et al., 2019; Keane et al., 2021)]. For instance, typically, studies use only a handful of test items and these materials are not properly counterbalanced over explanation conditions. Here, we try to rectify these deficits (indeed, we explicitly demonstrate how material-sets can matter).

So, the study presented participants with 32 test-images from the ImageNet dataset (i.e., 24 misclassifications, with 8 "fillers" that were correct classifications for attention checks) and were asked to make classification-correctness and explanation-helpfulness judgements of these items presented alongside one of the two explanation-types (NoBox or Box; i.e., no CCR or CCR). The 24 misclassifications were randomly divided into two material sets (A-set and B-set) to counterbalance the experiment; so, (a) one group (N=82) received the A-set with example-only explanations (NoBox) and the B-set with example-plus-CCR explanations (Box) and (b) the other group (N=81) received the A-set with example-plus-CCR explanations (Box) and the B-set with example-only explanations (NoBox; see Fig. 4 for examples). The statistical analysis then collapsed across these counterbalanced groups controlling for the effects of the material-set. This consideration of material-sets is not just a "statistical nicety", as any given user will only see a sample of these images, a good explanation strategy needs to operate successfully over different image-samples. Hence, user studies need to control for this issue by using different, reasonably-sized and randomly-sampled material-sets.

In summary, the study examined people's responses to the two post-hoc explanation-types, example-based explanation-only v example-based explanation-plus-CCR (i.e., NoBox v Box) over 24 image-misclassifications (i.e., 12 items for each type). The counterbalanced groups allowed us to test both explanation-types used for the same given image. As we shall see, that design also allowed us to examine material-set effects within the counterbalanced group (i.e., A-set v B-set)

### 5.1 METHOD

**Participants.** Participants (N=163) were recruited on the Prolific crowdsourcing site (www.prolific.co). All were aged over 18, native English speakers and lived in the USA, UK, or Ireland. Participants were paid £7.50/hr for their participation, which totalled £319.8. This N was chosen based on a power analysis for a low effect-size; this size was chosen because we anticipated the addition of CCR boxes would have a quite nuanced effect over just explanation-by-example due to it already being heavily preferred by users (Jeyakumar et al., 2020). This study passed ethics review of the institution (ref. after anonymous review).

**Materials.** Twenty-four misclassifications were randomly sampled from the CNN using ImageNet. These were actual test-image errors when the classification label differed from the ground truth. The twin-system method was applied to each prediction to find a nearest-neighbor example-explanation, and the CCR in the item (shown as a Box; see Fig. 5). The materials were randomly assigned to two different sets (A-set and B-set) and counterbalanced (as described earlier). Importantly, the sampling constrained the images to be both varied and those involving classes people could easily understand (e.g., snail, lemon, etc.).

**Procedure & Measures.** After being told the system was a program that "learned" to classify objects in images, people were told they would be shown several examples of its classifications (see

Appendix E for user study questionnaire). Their task was to rate the correctness (i.e., the question was "The program's labelling of the image is correct.") and helpfulness (i.e., the question was "The explanation helps me understand the program's labelling of the object in the picture.") of the presented classification on a 5-point Likert-scale from "I disagree strongly" (1) to "I agree strongly" (5). Each participant was shown 24 misclassifications (12 NoBox and 12 Box explanations) along with 8 filler items that were all correct classifications appearing every fourth question for attention checks. The 24 incorrect items were randomly re-ordered for each person. A debriefing on the rationale and background to the study was provided after testing.

## 5.2 RESULTS AND DISCUSSION

**Correctness & Helpfulness.** On average, people perceived the misclassifications as being equally incorrect/correct for both explanation types; overall, the mean correctness ratings for both explanation-types were the same, NoBox (M=1.85) and Box (M=1.85) and not significantly different (using a paired t-test, $t(162)$ = -0.018, 1-tailed, $p > 0.05$). So, people's perception of the correctness of misclassifications when given either of the explanation-types appear to be essentially identical. However, this analysis masks an important difference when ratings are broken out into the two material-sets. The items in the B-set that received the example-plus-CCR explanation (Box; M=1.93) were rated as less-incorrect (more correct) than their equivalent items in Set-A (M=1.77); this difference between B-set-Box and A-set-Box was statistically reliable, $t(161)$ = 2.15, $p = 0.03$, 2-tailed using a two-sample t-test. No other pairwise comparisons were statistically significant. This result suggests that image-explanations showing CCRs (as a boxed outline) impact people's perception of correctness of the misclassification, but only for certain items. We return to a more detailed analysis of this effect below. On helpfulness, both explanation-types were rated positively and equivalent, NoBox (M=3.16) and Box (M=3.11); using a paired t-test, $t(161)$ = -1.60, 1-tailed, $p = 0.06$. So, the two explanation options are equally helpful to users. Interestingly, for this measure, both material-sets produced almost identical ratings for both explanation-types and no other pair-wise comparisons are statistically different.

**Right\* & Wrong Misclassifications.** We were puzzled by the difference in correctness ratings for the example-plus-CCR (Box) conditions found in the B-set relative to the A-set. What is it about the B-set that produces this effect? In a follow-up analysis, we discovered that, in both material-sets there were ambiguous materials that people consistently rated as more correct (i.e., mean >2) even though the ground-truth identified these items as incorrect (3/12 in the A-set and 4/12 in the B-set). So, we partitioned the items into two new categories, namely "Right\*"-items (that people rated as more "correct", even though they were incorrect classifications) and Wrong-items (that people confidently rated as incorrect, when they were incorrect) and then re-analyzed the data for each material-set (n.b., we add an asterisk to "Right\*" on purpose to signal they are not really "Right"). We also verified this partitioning by clustering the material means (using k-means) and found that the data consistently forms these two groups across 500 iterations. Figures 4(A) and 4(B) show that

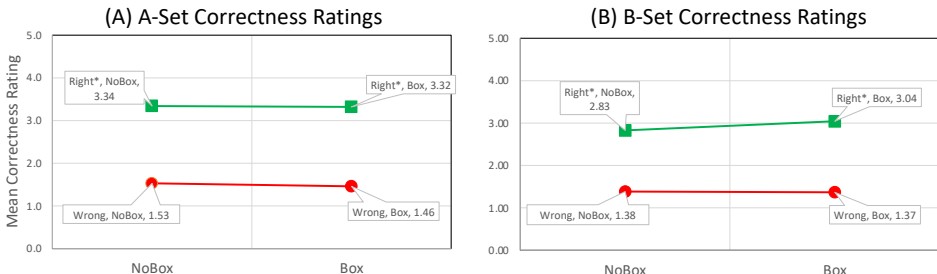

Figure 4: Expt. 3 results: Correctness ratings for two material-sets (A and B) broken out by "Right\*" and Wrong classifications for the two explanation-types, NoBox (example-only) and Box (example-plus-CCR). In the B-Set, the Right\*-Box ratings (M=3.04) are reliably different to the Right\*-NoBox ratings (NoBox, M=2.83), reflecting people's performance on ambiguous items in that material-set.

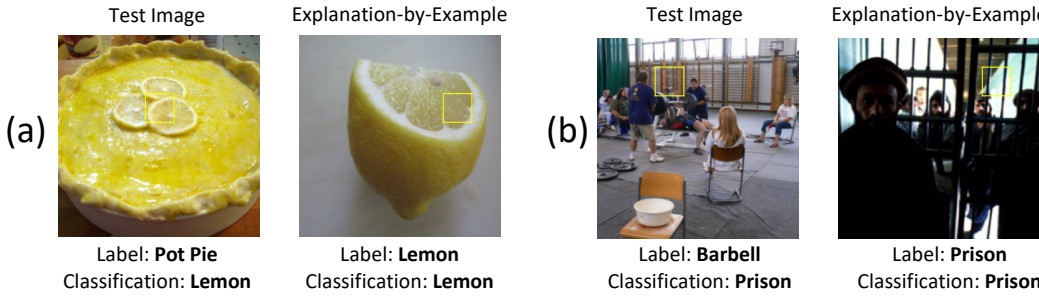

Figure 5: Expt. 3 Material Examples: (a) A "Pot Pie" is misclassified as "Lemon". The explanation shows the CCR in the image identified by the CNN. The explanation represents the training data and feature "used" in the CNN's classification. Glossed, the explanation says "I think this is a lemon, because it has a similar part to an image I saw before which I learned should be a lemon". (b) Another misclassification of a "Barbell" as a "Prison" in which the CNN picks up on the bars in the background reflecting the jail bars in a previous training image. Although anecdotal, an analysis of these two items shows that both have significantly higher helpfulness ratings when the CCR is given; t-tests show (a) NoBox 2.37 v Box 3.09, $p < 0.001$ and (b) NoBox 4.19 v Box 4.48, $p < 0.05$.

the "Right*" misclassifications are rated significantly higher than the Wrong ones, irrespective of the material-set (A or B) or the presented explanations (NoBox or Box; all $p < 0.05$ using t-tests). No other differences between conditions are statistically different, except for one involving the B-set; namely, the correctness rating for the Right*-misclassifications with example-plus-CCR explanation (Box, M=3.04) is reliably higher than that for the example-only explanation (NoBox, M=2.83), $t(162) = 1.8$, $p = 0.036$, using 1-tailed, two-sample t-test. Note that while this difference appears small on a Likert scale, overall it constitutes a 5.25% difference in responses, which is a larger effect than that found in other studies (Goyal et al., 2019). What does this mean? This finding suggests that some ambiguous material items (that people think may be correct, though they are incorrect) are affected by being given an explanation showing CCRs. Fig. 5(a) shows one such example, a picture of a "Pot Pie" (which is decorated with lemons), which the CNN labelled as a "Lemon"; here, the Box explanation shows the NN (an image of a lemon) and a boxed region showing the "pulp of the lemon" as the feature that influenced the classification. Clearly, when people see this CCR explanation, it provides more information about the classification, leading them to rate it as more correct (which does not occur for the NoBox version). Though this account makes sense, because this analysis is ad-hoc, it should be treated with some care. Nonetheless, it does show that there are specific image-items that are influenced by the provision of CCR information (presumably, when there is confusion over whether the AI really got the prediction right or wrong). Notably, and more generally, these findings give us a sense of the psychological complexities of XAI; namely, that there can be interactions between specific image-items, the predictions about those items made by the model, and the explanation-type used to justify those predictions.

## 6 GENERAL DISCUSSION AND CONCLUSIONS

Post-hoc explanation-by-example is one of the most popular and successful explanation strategies in both user testing (Buçinca et al., 2020; Jeyakumar et al., 2020), and computational experiments (Kim et al., 2014; Kenny & Keane, 2021). This paper has reported comparative tests of several ways to augment this popular explanation method by showing the most important *parts* of the image used, what we call *Critical Classification Regions* (CCRs). Initially, computational tests demonstrated the optimal way to isolate these regions in the test image and training data. Subsequently, a carefully designed user study showed that explanations with and without CCRs are both equally helpful for understanding, but that for certain ambiguous images the provision of CCRs influence people's perception of correctness. For future work, the investigation of CCR's utility into contrastive explanations will be considered, as well as how to link multiple/bigger CCRs in the test image to a single training example.

ETHICS STATEMENT

It is clear that XAI has made considerable progress in "opening the black box" to begin to reflect the internal workings of these successful, but complex, deep learning models. Here, we have seen that an explanation-by-example strategy shows some promise in supporting this effort (Jeyakumar et al., 2020; Kenny et al., 2021; Chen et al., 2019). However, we should also not loose sight of the potentially negative societal impacts that may arise from these XAI solutions. Perhaps the main concern being that, any data which is presented for explanation may need to be anonymized to ensure privacy and ethical concerns (Jeyakumar et al., 2020), particularly in sensitive domains such as medicine and law.

There are also wider issues to be considered around ensuring that these automated explanations inform without misleading end-users about the systems. Since CCRs gives users the impression that "incorrect" classification's seem more correct, a concern could be put forth that they may actively mislead end-users. However, since ImageNet has "real world" images which inevitably contain several classes (e.g., the pot pie in Fig. 4 could be a "lemon" or "pot pie"), many different possible classifications are arguably correct. So, the CCR is not necessarily "misleading" users, but rather pointing more precisely to where the CNN is looking, and (thanks to the nearest neighbour CCR explanation), "why" it is looking there (our main contribution). When users see the CNN is looking at the "lemon" when it classifies the "pot pie" as a "lemon", although some may see this as incorrect, it is nevertheless acting in a way which is objectively "correct", so people's feelings of "correctness" increased compared to the "nobox" version (or put another way, they see it as "less incorrect"). In terms of use-cases for CCRs, they appear to be most useful at convincing people a CNN's classification is correct, when there are several possible classifications a CNN could make. This would likely be useful in domains with many possible outcomes for which humans are not totally expert – a multi-class classification problem – (e.g., radiology diagnosis, see Fig. 11).

REPRODUCIBILITY STATEMENT

Much effort has been invested to ensure this work is reproducible. For instance, we are providing a code library which can be installed to use the CCR method easily. This includes the twin-ensemble method from Section 2. Moreover, to reproduce the experiments, we have provided all the code used in the supplementary materials, and all the data pre-processing used in the Appendix sections. The user study is given in full detail also, which shows exactly what we showed users, should anyone wish to recreate the results of this paper.

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

## A AN EVALUATION OF EXPLANATION-BY-EXAMPLE METHODS

The core novelty of this paper is CCRs, but to generate these the first crucial step is to locate a pool of "good" nearest neighbors (NNs) to avoid searching the entire training data. Rather than proposing a novel method, we found four candidate explanation-by-example methods that could deliver such pools (Jeyakumar et al., 2020; Papernot & McDaniel, 2018; Hanawa et al., 2021; Kenny & Keane, 2019). As an aside contribution, we competitively tested these four methods to see which we would use in the main paper's experiments, this aside experiment is detailed here.

The CCR algorithms proposed in this paper rely on obtaining a "good" pool of nearest neighbors. That is, the pool obtained via some nearest neighbor algorithm should be linked to the CNN/test image in question we are explaining, easily applied to any dataset/domain (regardless of the dataset size or complexity), and practical in the "real world". This experiment compares four explanation-by-example techniques: namely, ExMatchina by Jeyakumar et al. (2020), DkNN by Papernot & McDaniel (2018), Grad-Cos-x [inspired by Hanawa et al. (2021)], and twin-systems by Kenny & Keane (2019). Following Hanawa et al. (2021), we use the sanity checks proposed by Adebayo et al. (2018) to determine which method is most sensitive to random changes in the network's parameters. If a given method is not sensitive to this, it is reasonable to assume that the model's parameters are not being used to generate the explanation, and hence the explanation (i.e., the NN pool) is not linked to the CNN. Recent similar tests found Grad-Cos to be a good technique (Hanawa et al., 2021), but it does not scale to larger datasets, and their tests did not consider the three other methods tested here. Our modified Grad-Cos, called Grad-Cos-x, uses the loss gradients for similarity comparison at the final feature extraction layer in a given instance, rather than the entire CNN's parameters as described by Hanawa et al. (2021). Other techniques [e.g., Influence functions by Koh & Liang (2017)] were omitted due to them being computationally infeasible on larger datasets like ImageNet.

## A.1 SETUP, DATASETS, AND EVALUATION METRICS

Here, a pre-trained CNN $f$ is considered alongside four different randomized variants $f_{rand}$ (see Fig. 2). These variants randomized (1) the CNN's first layer, (2) half the convolutional layers, (3) the output linear layer, and (4) the entire CNN. The evaluation uses the first 500 instances from the test set of CIFAR-10 Krizhevsky et al. (2014) and FashionMNIST Xiao et al. (2017). An overlap set metric called the Szymkiewicz–Simpson coefficient is used to evaluate the NN pool. Specifically, a score of 0 means all the NNs are different, and 1 means the are all the same (i.e., the worst score). For each test image, a set of NNs ranging from 1-1000 is iteratively considered. Due to the complexity of ExMatchina (as the original paper suggested assessing similarity in the final convolutional layer rather than the penultimate linear layer, which increases the size of vectors for comparison $\times 49$), the evaluation is limited to relatively simple datasets. Furthermore, as the randomization of weights is susceptible to erroneous results, experiments are repeated five times with aggregated results shown in Fig. 2 along with standard error bars.

In addition, we also consider the "Agreement" metric proposed by Kenny & Keane (2021) to evaluate the methods. The method adjusts the training data labels to be what the CNN predicted them to be post-training, and then fits a $k$-NN classifier to the training data. The $k$-NN is then used to predict the test data and the amount these predictions overlap with the black-box CNN's predictions, is the final score. So, if all predictions are identical, an agreement score of 1.0 is recorded (which is the best score), if both systems disagree on every test prediction, a score of 0.0 is recorded (the worst score). Note this $k$-NN classifier uses $L_2$ distance for twin systems, but cosine similarity for the other three methods (as was proposed by the original authors). The intuition behind this evaluation metric is that the $k$-NN being used to explain the black box CNN should exactly match the CNN's predictions on test data, otherwise the NNs are not a faithful abstraction of the CNN function we are explaining.

Lastly, we were also inspired to use Spearman's Rank correlation to evaluate the NN pool found as suggested by Hanawa et al. (2021). The distance of the query to all NNs is recorded, and this is evaluated against the new randomized NNs. A score close to 0 means the NN pool is very different after weight randomization (as we want), divergent scores mean the opposite.

## A.2 RESULTS

Fig. 6 reports the results of the supplementary experiment. Overall, the results show that twin-systems are the best method across all tests. Fig. 6(A) shows the agreement metric with a perfect score of 1.0 for twin-systems, with DkNN, ExMatchina, and Grad-Cos-x doing worse in that order. Next, Fig. 6(B) shows the Szymkiewicz–Simpson coefficient, with twin-systems again the best method, with most of its NNs changing distance after the weight randomization, compared to the other three methods. Lastly, Fig. 6(C) shows results of the Spearman's Rank test, with twin-systems and Grad-Cos-x being the best methods, with less divergence in the former than the latter (indicating again that twin-systems are the best method).

Why does this happen? As DkNN and ExMatchina do not use weights in the output linear layer, their weight randomization in the $f_{rand}$ variant which randomized the weights in the last linear layer produced no difference. So aggregating the results of all tests showed DkNN and ExMatchina to perform much worse overall. Given the better performance of twin-systems and its claimed psychological validity Kenny et al. (2021), Expt. 1-3 focused on this method. Lastly, it is also worth noting that Grad-Cox-x requires testing labels to derived NNs, a constraint twin-systems do not have. Hence, although Grad-Cos-x performed well in these tests, we did not consider it, as we wanted a method which didn't require testing labels to be known *pre-hoc*, partly so it could also be used for self-supervised learning in future research.

## A.3 ARCHITECTURES

The architecture used in the Appendix A experiments is show in table 1. For CIFAR-10, the model's first Conv layer expected 3 dimensions (for color) rather than 1 on FashionMNIST.

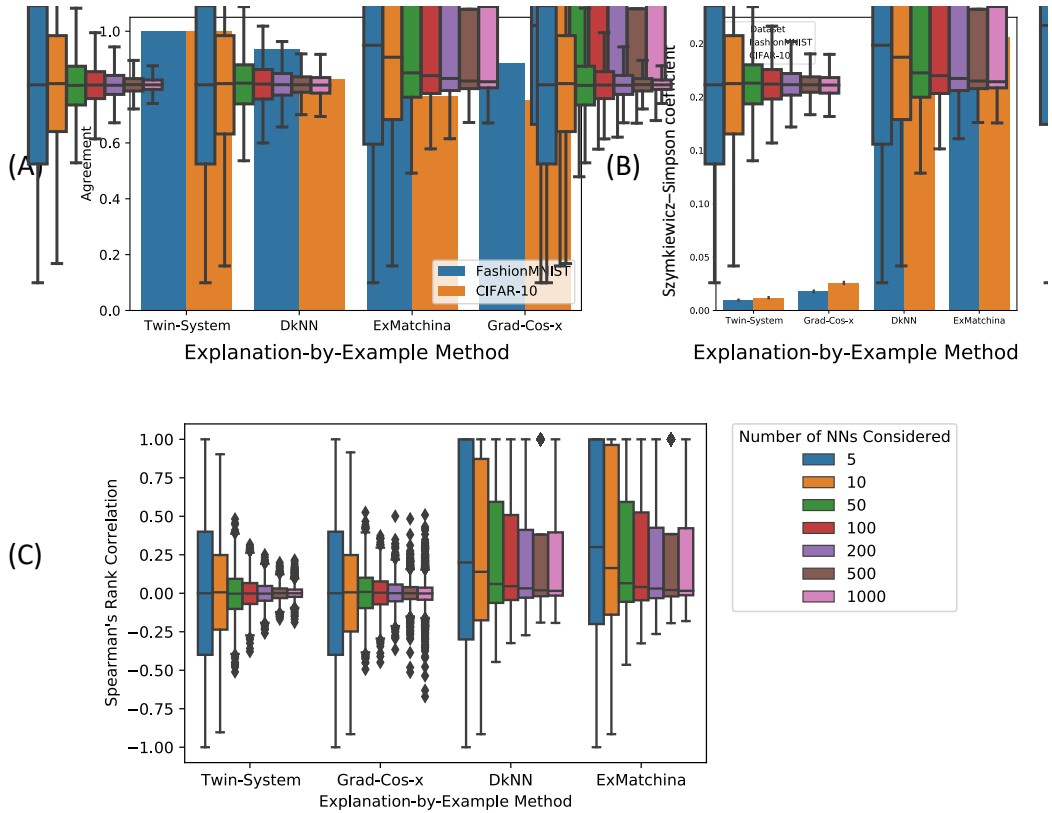

Figure 6: Experimental Results: (A) The agreement metric shows twins get a perfect score of 1.0, with DkNN performing second best. (B/C) The different layers of CNNs trained on CIFAR-10 and FashionMNIST are randomized to determine the sensitivity of the method to model parameters. (B) The Szymkiewicz–Simpson coefficient overlap set score shows how many NNs retrieved by the randomized CNN overlap with those from the non-randomized CNN, and the fully aggregated results from all tests are shown for both datasets in a bar plot. (C) Spearman's Rank shows a box plot with twins being the best method, with concentrated scores close to zero. Results show that overall twin-systems and Grad-Cos-x are the most sensitive across all layers (note the varying scales on the y-axis). Standard Error bars are shown in (B).

## A.4 TRAINING HYPERPARAMTERS

Both models for CIFAR-10 and FashionMNIST were trained the same way.

Adam optimizer was used with a learning rate of 0.01. Learning rate was decreased to 90% of it's previous value every epoch. The networks were trained for 20 epochs.

FashionMNIST used data augmentation by padding with 4, random resized cropping 28, random horizontal flips (p=0.5), color jitter (0.1 for brightness, contrast, saturation, and hue) in Pytorch. The network accuracy on the test data was 90.39%.

CIFAR-10 was the same except that random resized cropping was done at 32 pixels. It's accuracy on the testing data was 74.3%.

## B TWIN-SYSTEMS DETAILS

As it is the explanation-by-example method used in this paper, we present a brief summary of twin-systems here if it interests the reader. Twin-systems work by considering the final latent representation in an ANN (usually the penultimate layer), and fitting a $k$-NN to the training data's represen-

Table 1: The CNN architecture used to train models in Appendix A.

| Expt 1. CNN | |
|---|---|
| Layer | Layer Parameters |
| Conv2d
BatchNorm2d
ReLU | 8 filters, (5x5), (1x1), padding=2 |
| Conv2d
BatchNorm2d
ReLU | 16 filters, (5x5), (2x2), padding=2 |
| Conv2d
BatchNorm2d
ReLU | 32 filters, (5x5), (1x1), padding=2 |
| Conv2d
BatchNorm2d
ReLU | 64 filters, (3x3), (2x2), padding=2 |
| Conv2d
BatchNorm2d
ReLU | 128 filters, (3x3), (1x1), padding=1 |
| GAP | |
| Linear
SoftMax | 128, 10 |

tation at this layer, similar to Papernot & McDaniel (2018). However, twins are different because they weight these activations with feature-weighting methods. Specifically, a method known as Contributions Oriented Local Explanations (COLE) is typically used for the feature weighting.

What makes COLE interesting is that it ignores features which did not contribute to a classification when searching for nearest neighbors, and emphasises those that did, which allows $k$-NN to find better explanatory training instances. This approach helps ensure that the nearest-neighbor found is predicted in the same class as the test instance, and that they both use similar features in their classifications (Kenny & Keane, 2021).

The weighting vector $\vec{c}$ of some input to the network is given by:

$$\vec{c} = \left\langle x_1 \cdot \frac{\partial \hat{y}}{\partial x_1}, x_2 \cdot \frac{\partial \hat{y}}{\partial x_2} ... x_n \cdot \frac{\partial \hat{y}}{\partial x_n} \right\rangle \tag{3}$$

were $\{x_i\}_{i=1}^n$ represents the extracted features in an ANN (note in this paper it is the penultimate layer before the SoftMax output classification layer), and $\hat{y}$ the predicted output neuron class. The contribution scores for all the training data were generated and then fit to an $k$-NN for the twin-system. For all testing data, the process is repeated and its vector $\vec{c}$ used to search for nearest neighbors with the $k$-NN twin.

## C    EXPERIMENT 2: TRAINING HYPERPARAMTERS

For the training ablation studies ResNet50 and ResNet34 was used. ImageNet experiments used the pre-trained ResNet50 model available on Pytorch. CUB-200 fine tuned ResNet34 with the following hyperparamters.

**CUB-200.**    Epochs 500. 1 GPU used. Number of workers was 2. Batch size 12. Data transformations during training were done with Pytorch transforms: RandomResizedCrop(224), RandomRotation(45), RandomHorizontalFlip(0.5), ColorJitter(brightness=0.126, saturation=0.5). A multiplicitive learning rate decay of 0.999 was used. The epoch with best test accuracy was used for the model, which was epoch 41.

**ImageNet.** This was a pre-trained model from PyTorch.

Standard training and testing splits were used for ImageNet, and CUB-200. However, as is standard practice, the validation data was used for ImageNet, since the test data labels are unavailable.

**Fine-Tuning.** When finetuning the CNNs, the exact same hyperparamters were used (and the same ones from the original ResNet paper for ImageNet), except that the learning rate was decreased by $10^{-1}$ from each methods initial learning rate when training the initial CNN model.

## D   COMPUTATIONAL COSTS

### D.1   HARDWARE

Appendix A experiment had both the FashionMNIST and CIFAR-10 models trained on a single Nvidia K80 GPU, 2vCPU @ 2.2GHz, and 13GB RAM. The experiments were subsequently run on MacBook Pro, processor 2.9 GHz Intel Core i5, memory 16 GB 2133 MHz LPDDR3.

Experiments 1-2 were run on a Dell R740XD with an Nvidia V100 (32GB) GPU: 256GB RAM. Storage of the dataset was on scratch storage 220TiB.

**Computational time for experiments.** The Appendix A experiment took $\approx$8hrs to run each dataset. Expt. 1 took approximately 48hrs and 6hrs to run for ImageNet and CUB-200, respectively, presuming the use of a single Nvidia V100 (32GB) GPU. Expt. 3 took 2 weeks to run the beta experiment on ImageNet, one day for the alpha, and 7 days for the comparative tests. CUB-200 by comparison took less than 2 days for all tests.

## E   ALGORITHM PSEUDO-CODE

---
**Algorithm 1** Latent-Based-CCR Algorithms
---
**Require:** $f(.)$; CNN
**Require:** $x$; Test Image
**Require:** $NN(.)$; Nearest Neighbor Algorithm (Twin-Systems Recommended)
**Require:** $k$; Size of Nearest Neighbor Pool (50 recommended)
**Require:** $g(.)$; CNN model up to the final convolutional layer
**Require:** $\alpha$; Alpha hyperparameter (5 recommended)
**Require:** $m(.)$; Activation map algorithm (e.g., CAM)

$\quad C_x \in \mathbb{R}^{(h,w,d)} \leftarrow g(x)$ $\qquad\qquad\qquad\qquad\qquad$ ▷ Get Convolutional Output
$\quad M_x \in \mathbb{R}^{(h,w)} \leftarrow m(G_x)$ $\qquad\qquad\qquad\qquad\qquad\qquad$ ▷ Get Activation Map
$\quad \{N_i\}_{i=1}^{k} \leftarrow NN(x)$ $\qquad\qquad\qquad\qquad\qquad$ ▷ Get Pool of $k$ Nearest Neighbors

$\quad$ Select the segment $C_{i,j} \in \mathbb{R}^{(1,1,d)}$ with the maximum saliency $M_{i,j}$ as the test image CCR $\omega_{test}$.

$\quad$**for** $n_i \in \{N_i\}_{i=1}^{k}$ **do**
$\quad\quad\quad C_x \in \mathbb{R}^{(h,w,d)} \leftarrow g(x)$
$\quad\quad$**for** $i$ in range $h$ **do**
$\quad\quad\quad\quad$**for** $j$ in range $w$ **do**
$\quad\quad\quad\quad\quad \omega_c \leftarrow C_{i,j} \in \mathbb{R}^{(1,1,d)}$
$\quad\quad\quad\quad\quad$ Record $L_2$ distance $l = \|\omega_c - \omega_{test}\|_2^2$
$\quad\quad\quad\quad$**end for**
$\quad\quad\quad$**end for**
$\quad$**end for**

$\quad$ Select the neighbor $n$ with segment $i, j$ which minimised Eq. (1).

---

---

**Algorithm 2** Superpixel-CCR Algorithm

---

**Require:** $f(.)$; ANN
**Require:** $x$; Test Image
**Require:** $NN(.)$; Nearest Neighbor Algorithm (Twin-Systems Recommended)
**Require:** $k$; Size of Nearest Neighbor Pool (50 recommended)
**Require:** $SA(.)$; Superpixel Algorithm (SLIC Recommended)
**Require:** $\beta$; Beta hyperparameter (infinity recommended)

$S_x \leftarrow SA(x)$           ▷ Get Superpixel Segments
$\{N_i\}_{i=1}^k \leftarrow NN(x)$        ▷ Get Pool of $k$ Nearest Neighbors

**for** $s_i \in S_x$ **do**
    Set all parts of $x! = s_i$ to 0
    Pass image though $f(.)$
    Record output logit $y_i$ of the neuron corresponding to the prediction of $f(x)$
    Modify $s_i$ by upsampling it to the $f$ input size whilst maintaining the aspect ratio
    Pass the upsampled $s_i$ though $f$
    Record the latent representation of the upsampled $s_i$ as $l_i$
**end for**

Select the segment $s_i$ with the maximum $l_i$ as the test image CCR $\omega_{test}$.

**for** $n_i \in \{N_i\}_{i=1}^k$ **do**
    $S_n \leftarrow SA(x)$
    **for** $s_i \in S_n$ **do**
        Set all parts of $n_i! = s_i$ to 0
        Pass image though $f(.)$
        Record output logit $m_i$ of the neuron corresponding to the prediction of $f(x)$
        Modify $s_i$ by upsampling it to the $f$ input size whilst maintaining the aspect ratio
        Pass the upsampled $s_i$ though $f$
        Record the latent representation of the upsampled $s_i$ as $l_i$
    **end for**
**end for**

Select the neighbor $n$ with segment $i$ which minimises Eq. (2).

---

## F    EXAMPLE EXPLANATIONS

Here more example explanations are showcased with correct classifications, and incorrect classifications. In addition, a misclassification with three CCRs and multiple NNs (rather than just one) to illustrate this explanation option is shown.

Fig. 7 shows six correct classifications on ImageNet by ResNet50, alongside an explanation for them. Firstly, the explanation comprises of a nearest neighbor from a pool of $n$ candidate cases retrieved (50 in our experiments), which alone is already considered a "good" explanation by non-experts (Jeyakumar et al., 2020; Kenny et al., 2021). However, the explanations go further by pinpointing a CCR in the test image which was "learned" from the NN. This type of explanation not only informs the user of what important feature was used in the explanation, but also from *where* it arose in the first place so it may be further contextualized.

Fig. 8 shows two incorrect classifications in ImageNet from our user study. The first image is a "Kimono" misclassified as a "Violin", were the CNN confused the pipe in the test image with a violin bow. Fig. 8b shows a "Hammer" misclassified as a "Shovel". Here the CNN saw similarity in the test image's wooden handle to a previous training image of a "Shovel", and hence classified the image as a shovel.

Fig. 9 shows another two incorrect classification in ImageNet from our user study. Fig. 9a shows an "Acoustic Guitar" which is misclassified as an "Electric Guitar". The CNN conflates the fretboard as being indicative to an electric guitar, but it is also important to an acoustic guitar, and hence the misclassificaiton arises. Fig. 9b shows a misclassificaiton of a "Flute" as a "Horizontal Bar". The visual similarity of the horizontal bar in the NN image is confused with the wooden flute in the test image.

Fig. 10 shows another way of visualizing CCRs with multiple NNs shown. Specifically, the three most salient CCRs are shown alongside the three closest representations of them in the pool of NNs retrieved. Interestingly, in Fig. 10, the third CCR seems to be confused between a Leopard's tail and leg, possibly contributing to the misclassification.

Fig. 11 shows another example of using three latent-based CCRs for the correct diagnosis of Covid-19 in a patient's x-ray.

Fig. 12 shows another example of using SP-CCRs to explain misclassifications on ImageNet. (A) A "Knot" and (B) "Rifle" are misclassified by ResNet50 apparently due to bias in the CNN by associating, what could be called, "wooden background", and "snowy background" with the respective classes.

## G    USER STUDY SCREENSHOTS

For complete transparency on the user study design, we include screenshots (anonymous during review) for the reviewer's convenience. In this section exactly what all users of our study saw is shown, page by page, in the exact order. The bulk of the questions are omitted for space and clarity, but all original images used in the study may be acquired from our supplementary folder for this paper.

Or, if you prefer, you may sample your own materials to recreate the study.

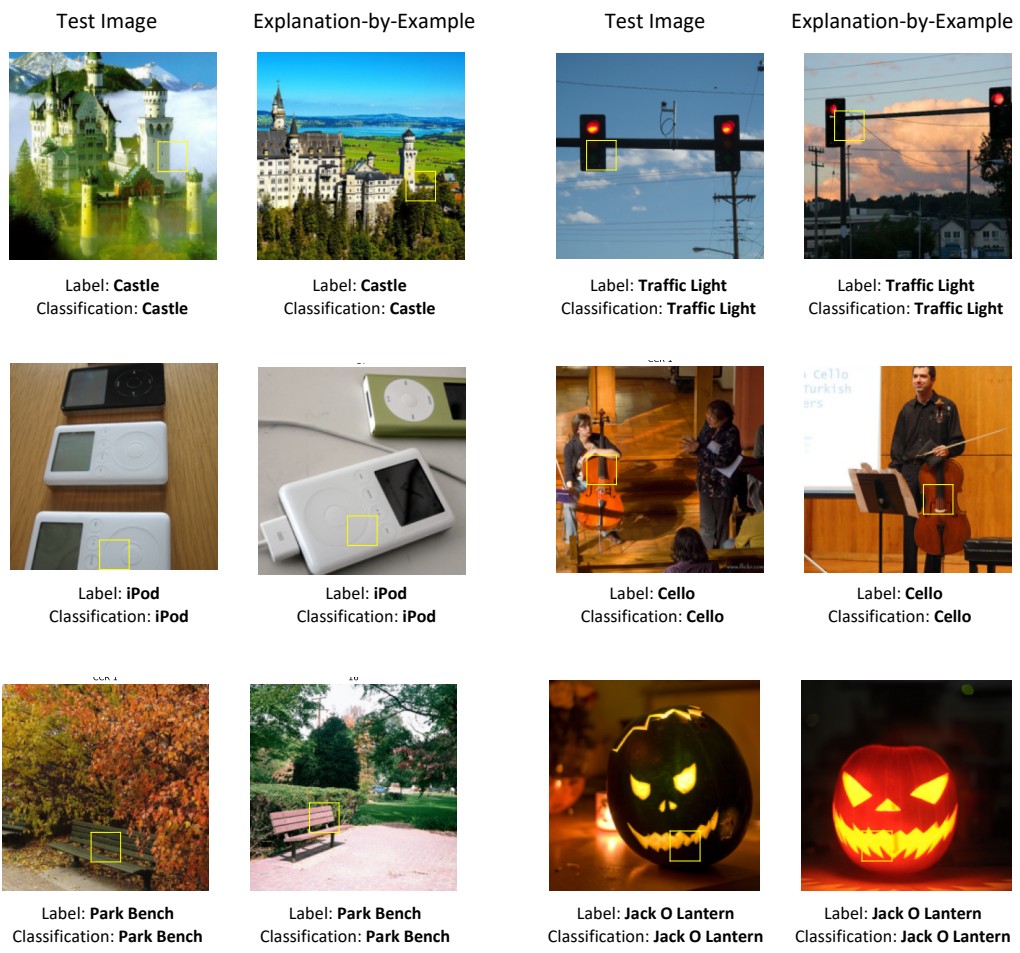

Figure 7: Correct examples: Starting from the top left (and going in reading order), we see correct classifications and a NN explanation showing the CCR used in the classification. Namely, the images show correct classifications of a "Castle", "Traffic Light", "iPod", "Cello", "Park Bench", and "Jack O Lantern"

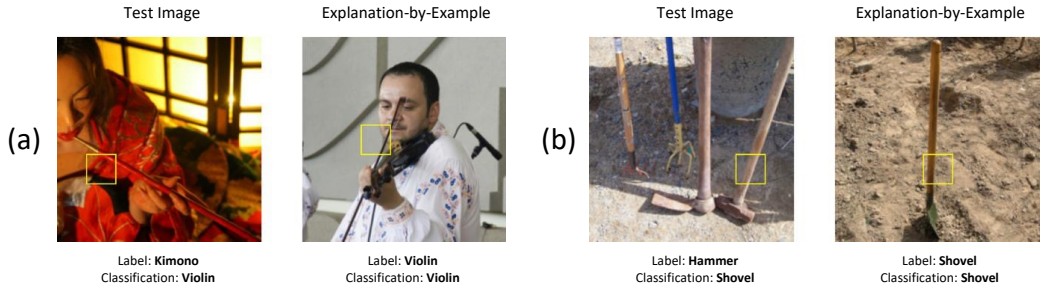

Figure 8: Incorrect examples: (a) A "Kimono" is misclassified as a "Violin", the CCR shows the CNN confused the pipe in the test image as the violinist's bow. (b) A "Hammer" is misclassified as a "Shovel", the CCR shows the CNN learned to focus on the wooden handle of shovels when classifying them, which it partly learned from the training image shown.

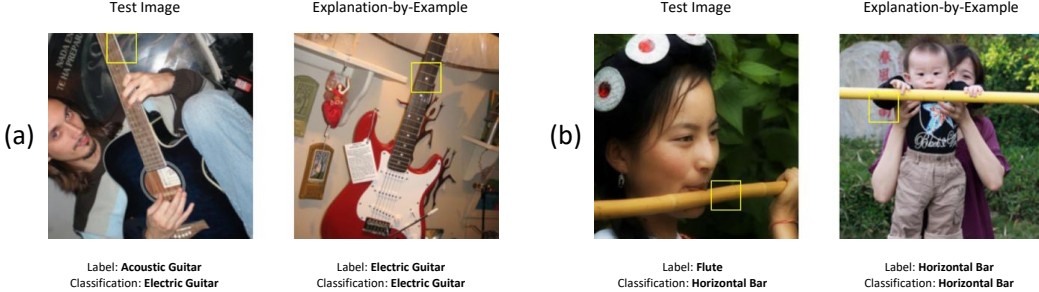

Figure 9: More Incorrect examples: (a) An "Acoustic Guitar" is misclassified as an "Electric Guitar", the CCR shows the CNN has learned to associate the guitar's fretboard with electric guitars, and neglected the rest of the image when classifying the test image. (b) A "Flute" is misclassified as a "Horizontal Bar", the CCR shows the CNN seems to have focused on the qualitative similarity between the horizontal bamboo bar in the training image, and the bamboo flute in the test image.

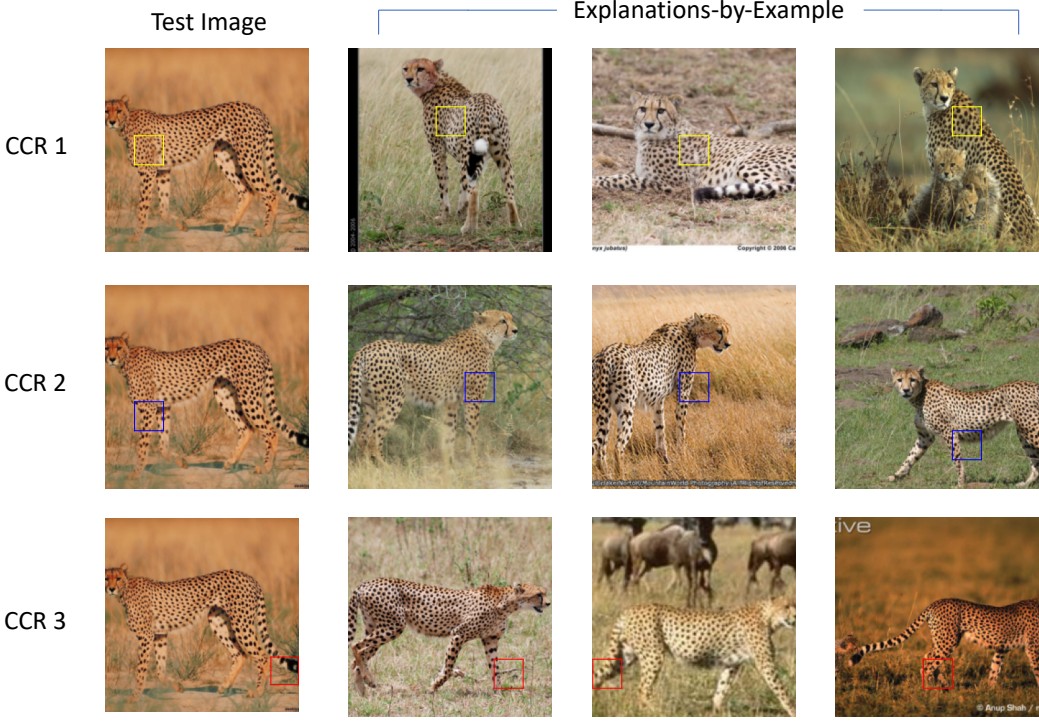

Figure 10: Multiple CCRs: An incorrect classification of a "Leopard" as a "Cheetah". Three of the most salient CCRs are shown for the test image, alongside their three closest representations in the pool of NNs.

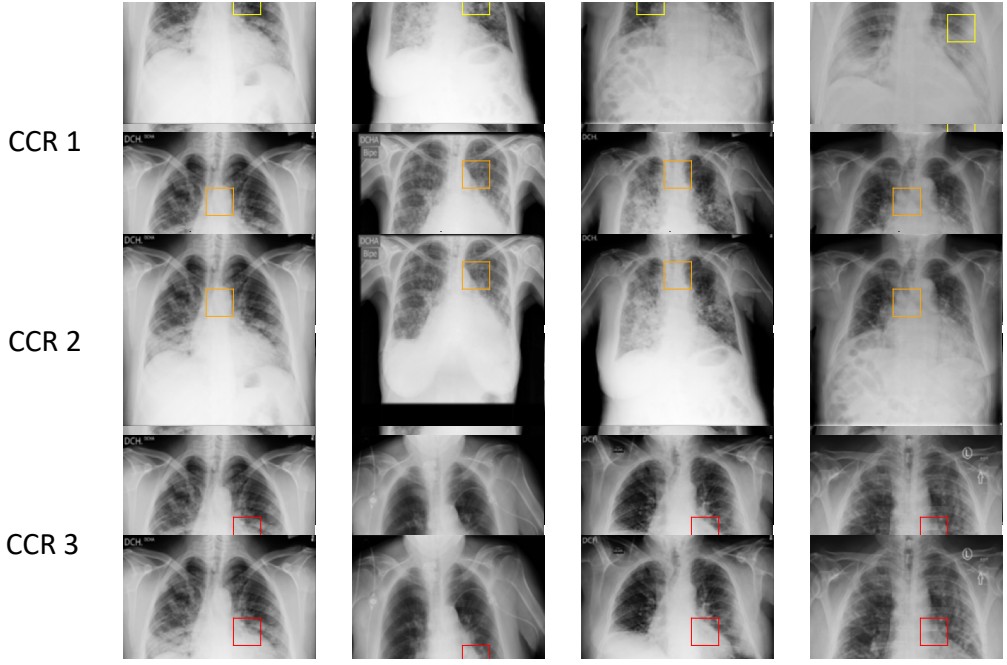

CCR 1

CCR 2

CCR 3

Figure 11: Multiple CCRs: A correct classification of "Covid-19". Three of the most salient CCRs are shown for the test image, alongside their three closest representations in the pool of NNs.

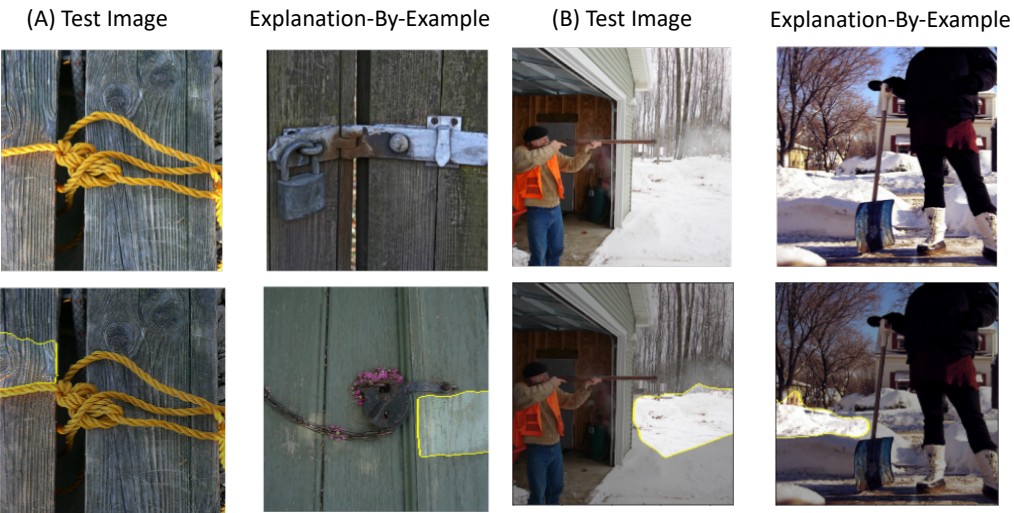

Figure 12: Incorrect ImageNet Classifications: SP-CCRs are used to explain two misclassification (A) A "Knot" is misclassified as a "Padlock", were the CNN has learned to associate a "wooden background" feature with the predicted class, causing a bias. (B) A "Rifle" is misclassified as a "Shovel", where a bias in the CNN has learned to associate a "snowy background" feature with the class "Shovel".

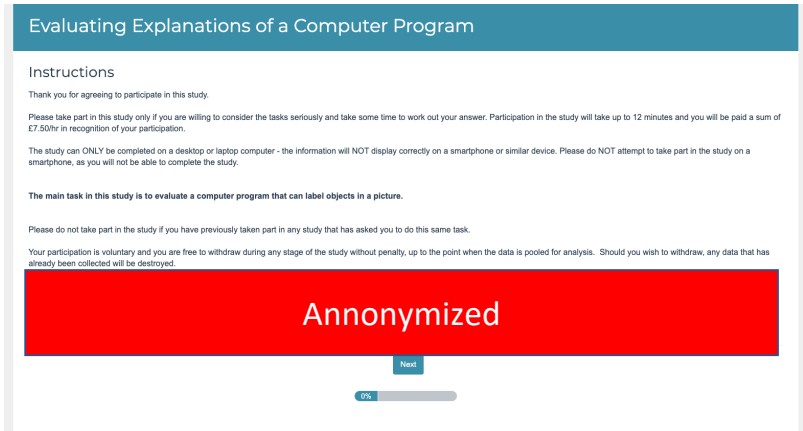

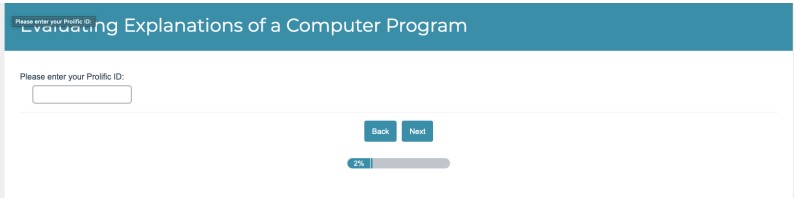

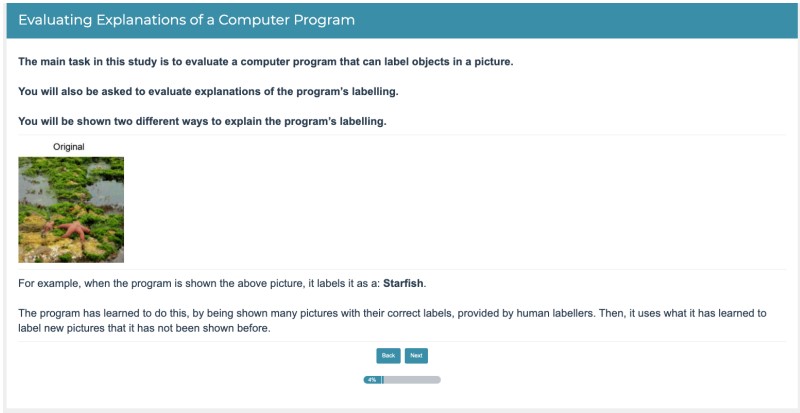

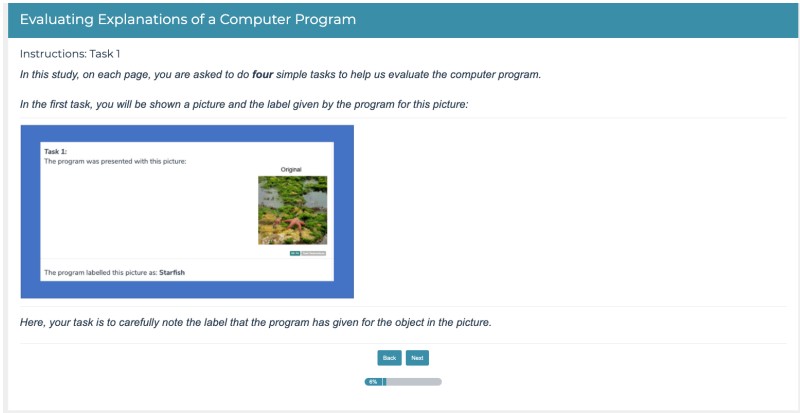

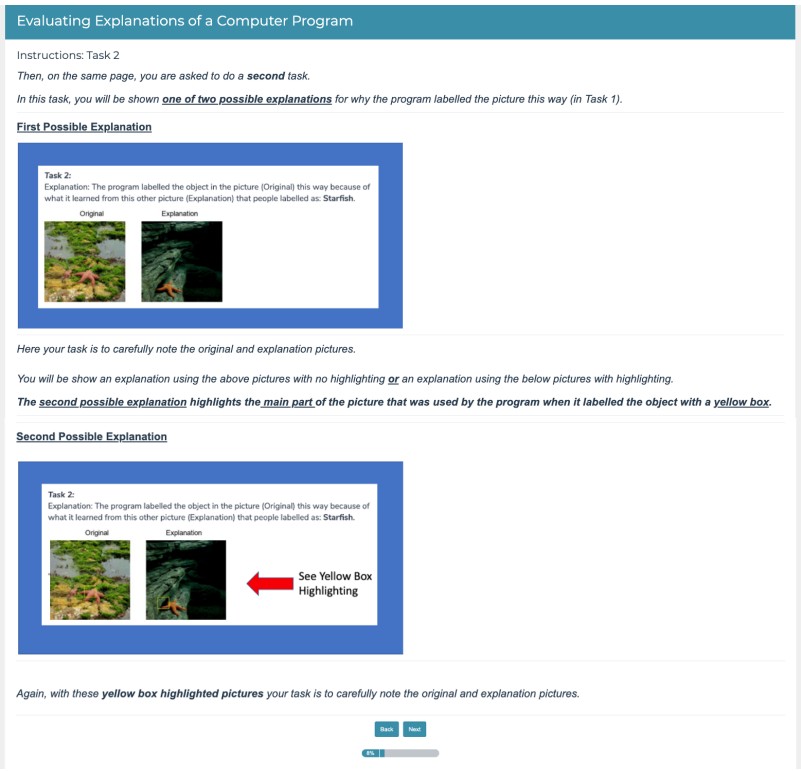

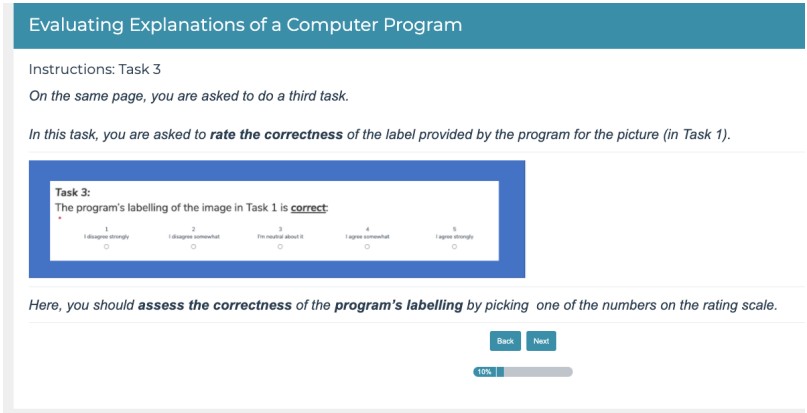

## Evaluating Explanations of a Computer Program

Instructions: Task 4

*Finally, on the same page you are asked to do a fourth task.*

*In this task you will be asked to **rate the explanation** (in Task 2).*

*That is, to evaluate whether the explanation helps you to **understand** the program's labelling*

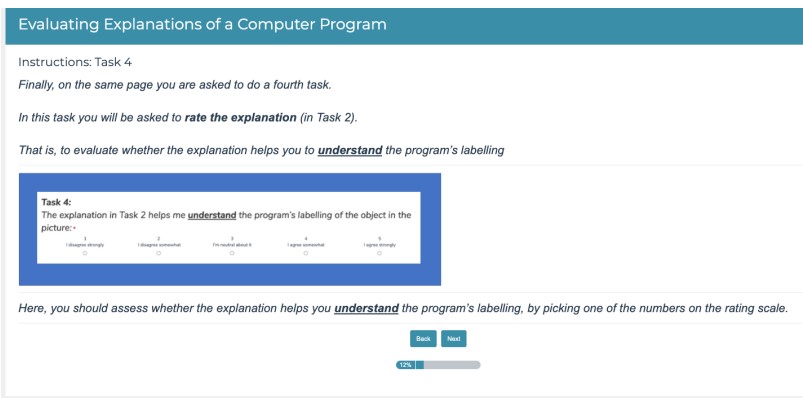

*Here, you should assess whether the explanation helps you **understand** the program's labelling, by picking one of the numbers on the rating scale.*

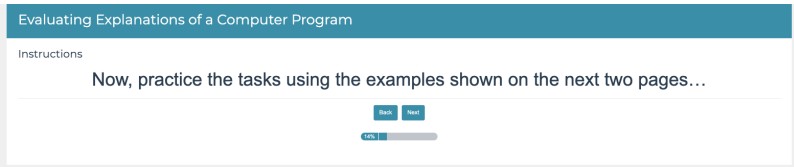

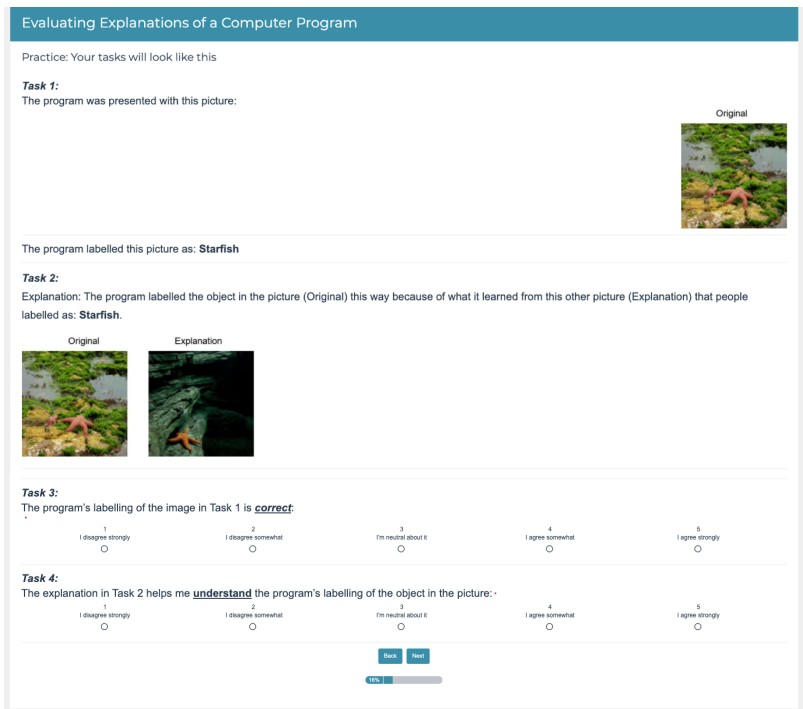

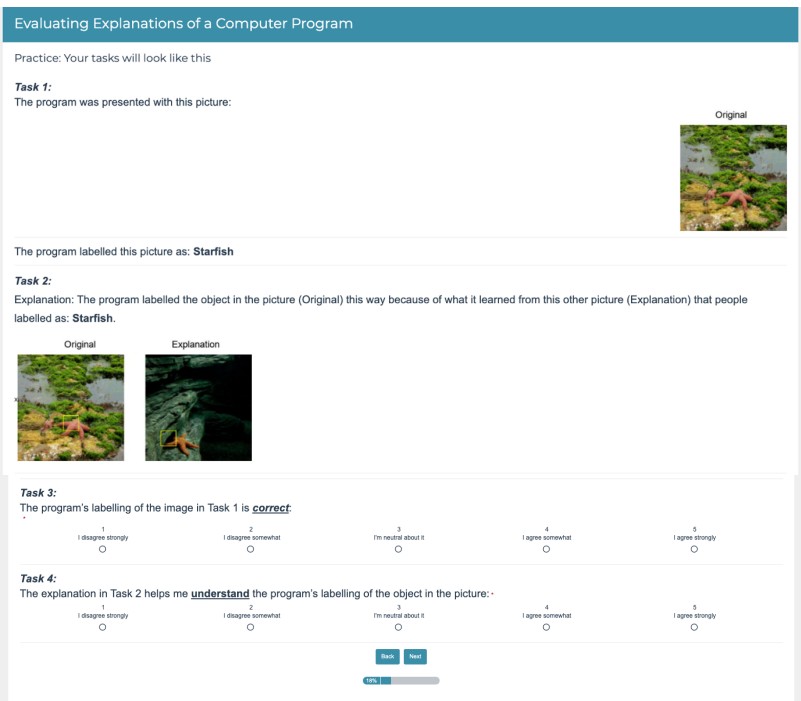

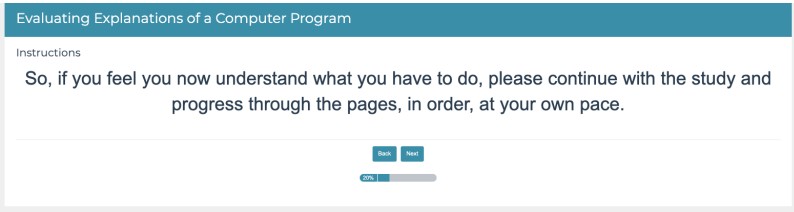

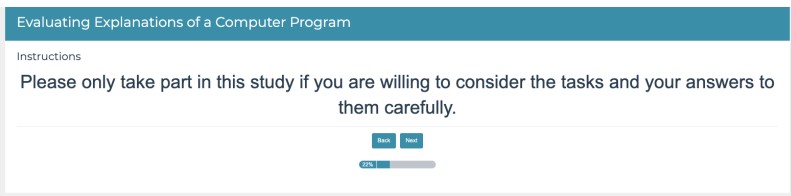

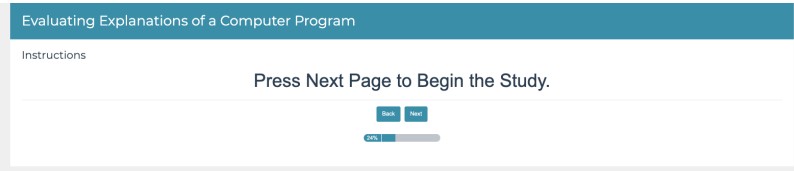

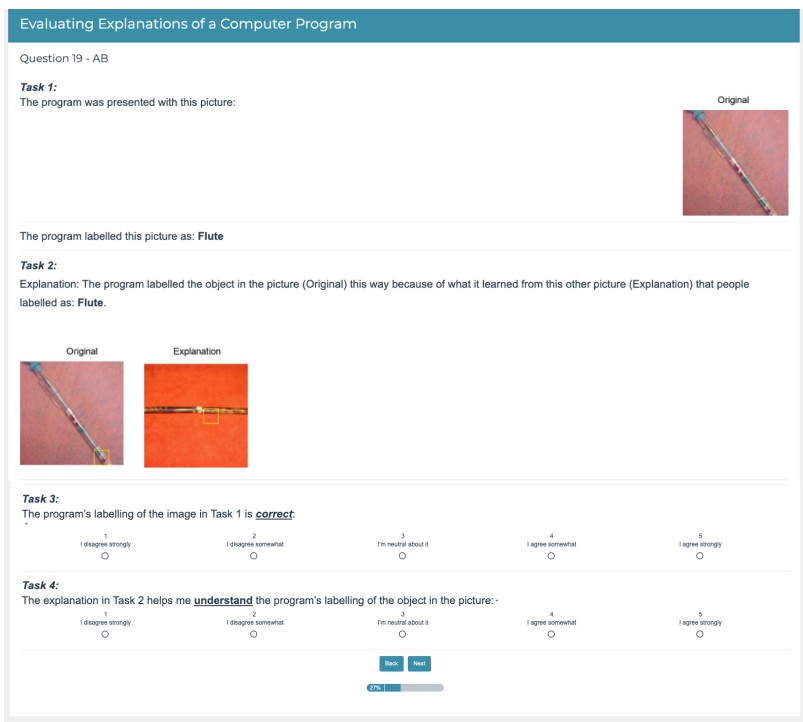

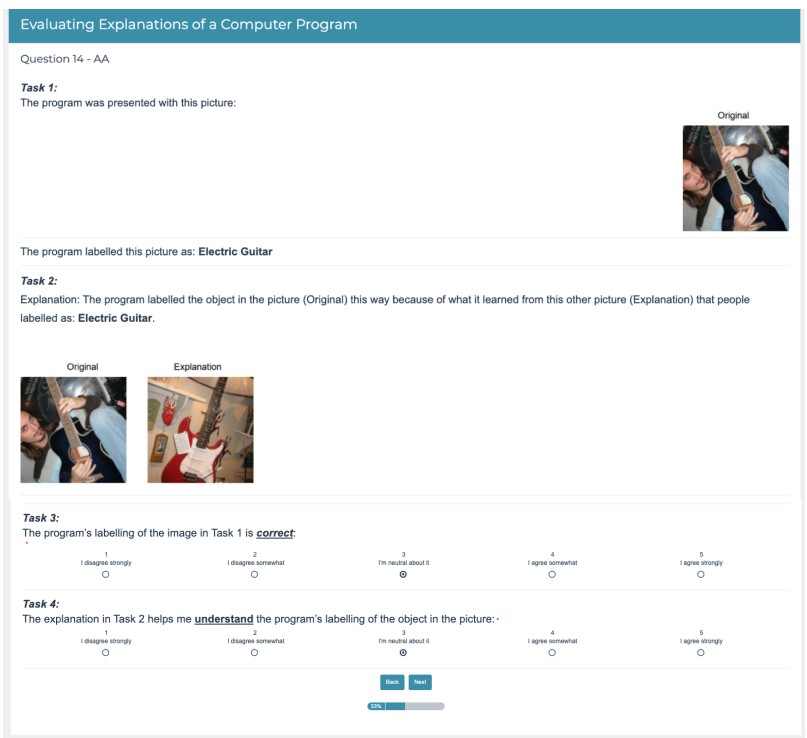

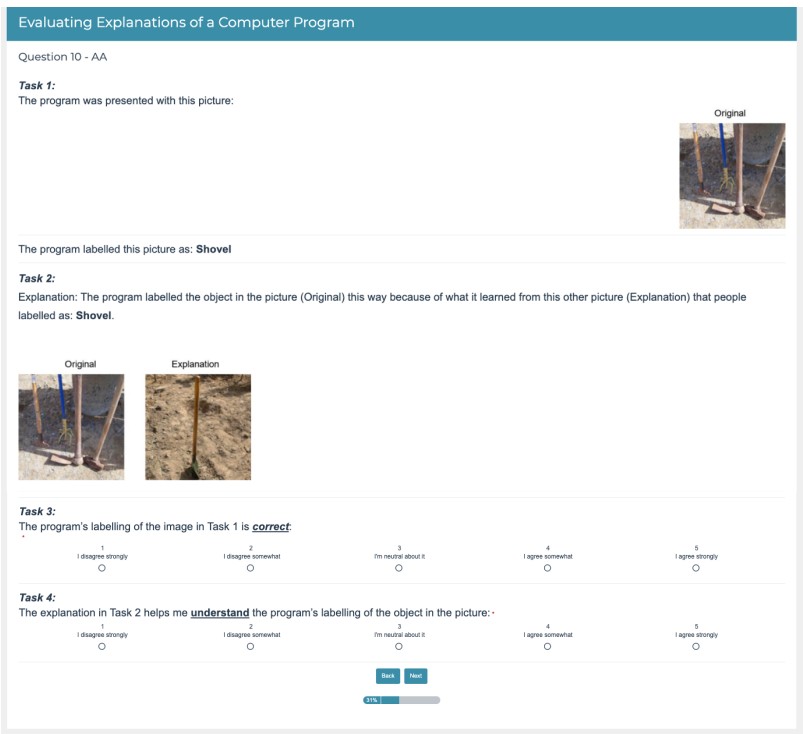

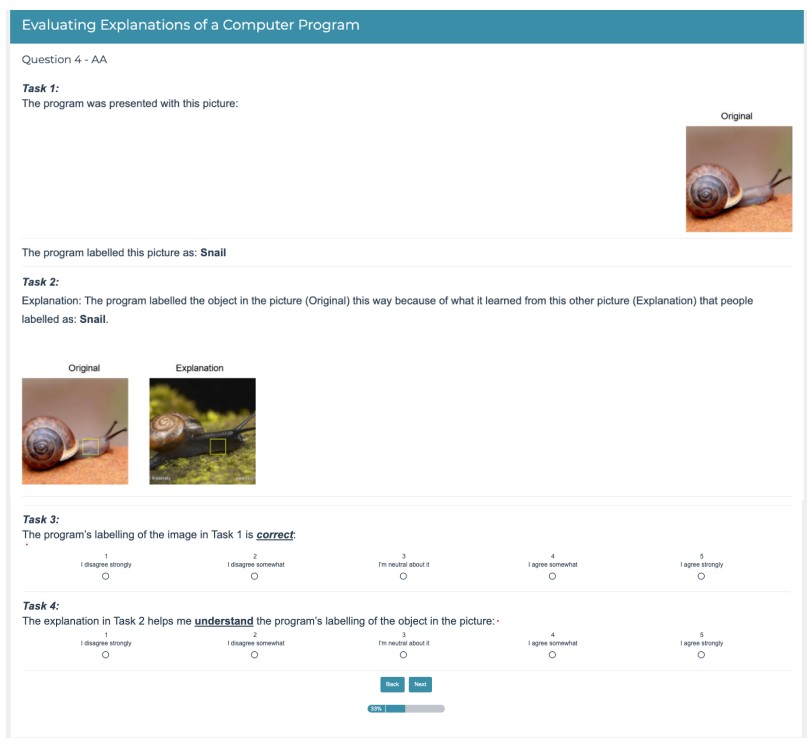

Continue for 32 Questions…. See main paper for study layout.

All original materials may be sampled from ImageNet in the supplement.

Recall there are two versions of the study, one were material Set-A has a CCR explanation, and Set-B does not, and then the other study which does the opposite. All incorrect classifications are randomized.

Correctly classified (e.g., the snail above) materials were spaced out every 4 questions and were not randomized (i.e., question 4, 8, 12… 32)

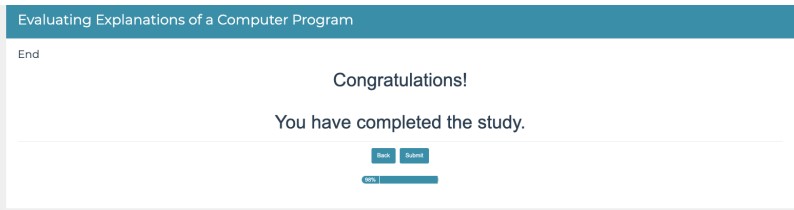

## Debrief Page

This study is being conducted to determine the effect of the use of explanations for the outputs of computer programs. Your responses will be used to compare the perceived usefulness of computer program explanations such as this used here.

**Annonymized**

Thank you for taking our survey. Your response is very important to us.

If you have any further questions about the study, or would like to access a summary of the findings, please contact:

**Annonymized**

**You must enter this code on the Prolific site.**

**If after 30 seconds you are not automatically redirected to the Prolific completion page, please click the link below. You must be redirected or click the link to prove that you have completed this survey.**

**Annonymized**

