# OpenReview forum: "Advancing Nearest Neighbor Explanation-by-Example with Critical Classification Regions"
_ICLR.cc/2022/Conference — ICLR 2022 Submitted_

### Official Review · Reviewer_sNwm · 2021-10-29

**Correctness:** 3
**Technical Novelty And Significance:** 3
**Empirical Novelty And Significance:** 4
**Recommendation:** 6
**Confidence:** 4

**Main Review:**

The paper presents a nice advancement of the state-of-the-art in XAI. The proposed method is partially incremental as it is based on the composition of existing approaches for the various steps, and their composition happens in a rather simple way. However, I believe that it brings innovation to the field as it seems effective from the evaluation. The weak points are listed in the following. First, the twin-system that is at the basis of the proposal is not properly presented and illustrated. It is impossible to fully understand the paper without reading the papers presenting the twin-system. Second, Experiment 1 is biased because the way adopted to test the importance of the parts of the image, i.e., by image occlusion, is exactly the same used by SP-CCRS. Thus this method has a clear advantage over the others. The authors should adopt a validation fair for both proposals or test them into separate settings. Third, the paper lacks for comparison with a method like the one of Been Kim or Cynthia Rudin. This statement is true for Experiments 1, 2 but also for Experiments 3. In this last case, in fact, I suppose that the examples are extracted with the twin-system. What about examples retrieved by the proposals of Been Kim or Cynthia Rudin? Also, a comparison against counterfactual explanations (see Yash Goyal), would have provided a better picture. Finally, the very interesting case study, unfortunately, does not bring the expected results. I suppose the way the questions are proposed to the users impacted the results as the explanation provided with CCR should be better than those having only examples. I suggest the authors repeat the experiments focusing only on images correctly classified.

Minor issues. I suggest avoiding using k for both the number of classes and the k of the NN algorithm.
The number k used in the experiments is not clearly stated as well as which are the different versions of kNN tested.

**Summary Of The Paper:**

The paper proposes a local model-specific post-hoc explanation method for image classifiers (CNN) returning as explanations examples suggesting the reasons for the classification together with subparts of the images named critical regions common to the test instance and the examples and responsible for the classification.

**Summary Of The Review:**

A very nice idea probably a bit incremental.
An experimentation not sound in some aspects.

---

> ### Author Response · Authors · 2021-11-18
> **Author(s) Response**
>
> We thank the reviewer for their positive and observant remarks, we answer all concerns and how we revised the manuscript:
> ***
>
> ### 1:  Twin-systems not properly presented
> Agreed, we omitted twin-systems from the main paper and moved it to Appendix B, along with a full description if it interests the reader. Note however that the CCR method will work with any explanation-by-example method.
>
> **Revision:** We moved twin-systems to Appendix B.
> ***
>
> ### 2:  Experiment 1 is biased
> This is a very astute observation.  Originally, we thought about occluding the test image CCR rather than everything else as the submitted version of the paper does (because as you note this is likely fairer). However, to our surprise, when we saw that CAM-CCRs actually did better in some circumstances, we decided to leave it. However, in the revision we have addressed your concern and replaced Expt. 1 with a new version which *both occludes* the CCRs (to be fairer as you suggest), *and includes* them with everything else occluded (what we did before), so we examine every possible avenue for completeness. In addition, we also included a comparison to LIME to see if diverges from SP-CCRs (as LIME is widely accepted as highly discriminatory). Note that due to the LIME comparison, we also occluded images with a “grey” background rather than a “black” one as before (because LIME does that), so, whilst the overall pattern of results is the same, they are slightly different now.  We hope the reviewer agrees this rewrite is a thorough and earnest response to their concern.
>
> **Revision:** Reran Expt. 1 with including/removing CCRs, with LIME comparison.
> ***
>
> ### 3:  The paper lacks for comparison with a method like the one of Been Kim or Cynthia Rudin.
> We are fans of Been Kim’s and Cynthia Rudin’s work and they partly inspired the CCR method. However, Rudin’s work is for **pre-hoc** interpretability, whereas ours is about **post-hoc** explanation (para. 5); so the underlying methods are very different, especially since ours considers superpixels, and theirs doesn’t. Moreover, despite 250+ citations and much follow-up work, [1] has not been shown to work on ImageNet, so unfortunately it seems computationally impractical here. Kim’s work mostly crosses ours with [2], but these are closer to global explanations than the local explanations we develop, so they cannot be compared in a faithful way to both methods. The other relevant work by Kim is [3], but it doesn’t highlight important regions like CCRs. Again, although we would like to compare these methods for additional baselines as the reviewer suggests, we are at a loss to know how it could be possible.
> ***
>
> ### 4:  A comparison against counterfactual explanations would be good
> We have other work on counterfactuals and are very conscious of Goyal’s excellent contributions. However, counterfactual explanations are very different to the factual ones we examine here; they demand very different mechanisms and elicit very different responses from users. Goyal’s algorithms find the minimal change to modify an instance into a predetermined counterfactual class whereas ours finds a factual build-up of evidence for why a classification occurred in the first place.  Again, although we would like to compare to this method for an additional baseline, we are at a loss to know how it could be possible, especially for our superpixel method which is so different. As an aside, we should also say that the Goyal method has also not been demonstrated to work on ImageNet, so unfortunately we are again doubtful it is feasible.
> ***
>
> ### 5:  Finally, the very interesting case study…
> Thank you for this observation! It has been observed in the XAI literature that explanations mostly matter when people encounter errors/misclassifications or something surprising happens (as was again seen in the current study). We think the questions asked of users were ok, and consistent with related work [4], what we discovered (which was not expected) was that there was a new class of error in certain query items (these ambiguous items), and these were the key occasions when the explanation mattered (recall there were no differences in the correct items). As such, we would argue that the results of the experiment deserve to be shared with the community as is, rather than ignoring them. However, as per your suggestion, we do expect that explanations for correct classifications will matter in more complex domains (e.g., radiology; see Fig. 11).
> ***
>
> ### 6:  Minor issues: k notation and value.
> Revision: Agreed, we removed this section and mentioned the k used in Section 2’s intro.
> ***
>
> [1]  Chen, C., et al., 2018. This looks like that
>
> [2] Ghorbani, A., et al., 2019. Towards automatic concept-based explanations.
>
> [3] Kim, B., et al., 2016. Examples are not enough, learn to criticize!
>
> [4] Kenny, E.M., et al., 2021. Explaining black-box classifiers using post-hoc explanations-by-example. *Artificial Intelligence*

---

### Official Review · Reviewer_emYe · 2021-11-01

**Correctness:** 2
**Technical Novelty And Significance:** 2
**Empirical Novelty And Significance:** 2
**Recommendation:** 3
**Confidence:** 3

**Main Review:**

- Overall the proposed idea is clear, I find its simplicity a good strength of the proposed method.
- I find the user-study presented in Section 5 adequate and with a proper level of depth.
- In addition, it seems that a code library related with the proposed method will be released after publication of the manuscript. This is always welcome from the reproducibility point of view.

My concerns with the manuscript are the following:

- The novelty of proposed proposed CCRs is reduced, and somewhat incremental. Moreover, it is only limited to explanation-by-example methods.

- To a good extent, the idea behind CCRs of highlighting the regions that link input images with training examples is similar to that of prototype-based explanations, especially the work from [Chen et al., 2019]. In this regard, proper positioning with respect to that family of methods would strengthen the manuscript.

- It is not clear to me what the value on the y-axis of Figure 2 (A & B) refers to. Is that the activation value of the logit related to the predicted class or the ground-truth class? how do you handle the cases when the prediction changes over the evaluation of different segments within the same image?

- Perhaps I missed something, but in Section 4, it seems that 30 versions of the original model are fine-tuned by gradually considering (adding) the superpixels of each image in the order of importance predicted by the explanation method. (Each time a segment is added, the model is fine-tuned. From this procedure, it seems that the evaluation is still more related towards assessing the relevance order in which the regions of the image are ranked by the explanation methods and not really the effect that specific training examples may have on performance.
- Also, I was wondering if these networks are not trained from scratch, then there is no guarantee that information present in the held-out segments is actually ignored since the model may still contain features related to then.

- Also, in Section 4, the manuscript explicitly links observations made in the experiment conducted therein with causality. Given the fact that the tested model is continuously modified on that experiment (none of the 30 fine-tuned resulting models is the same as the original model), I do not see how analyzing these different models can tell us something regarding causal properties of the original model.

- Proper validation of a method is not a contribution per se but a requirement in place for a method to be accepted/adopted by the community. There do not seem to be any novel protocols proposed in this regard, if that is indeed the  case, this should not be claimed as a novelty.

- Text and legends present in the plots from Figure 2 is not legible. I had difficulties reading it in digital version of the document. Please ensure the text have sufficient size to be readable when printed in standard A4 paper.

References
- Chen et al., This Looks Like That: Deep Learning for Interpretable Image Recognition
NeurIPS 2019.

**Summary Of The Paper:**

The manuscript proposes a method to extend explanation-by-example (aka. exemplar-based explanations) by highlighting the regions that links the test image with example(s) provided as part of the explanation. Towards this goal, different methods to compute Critical Classification Regions (CCR) are proposed where image regions/pixels linking the input with the explanation examples are highlighted.

The proposed method is validated on the CUB-200 and ImageNet dataset. This is complemented with a user-study.


**Summary Of The Review:**

While there is novelty in the proposed method,I find it quite limited. Moreover, at this point, I have several doubts regarding the quantitative evaluation, i.e. Sections 3 and 4, of the proposed method (please see my review). Perhaps it is a problem of clarity, but is something to be addressed if the manuscript is to be published as a solid piece of scientific work.

---

> ### Author Response · Authors · 2021-11-18
> **Author(s) Rebuttal**
>
> We thank the reviewer for taking the time to review our work, we respond to all your concerns below in order and how the manuscript was revised to address them:
> ***
> ### 1. Limited Novelty.
> For 20+ years, case-based reasoning has argued that *k*-NN’s naturally explain predictions using explanation-by-example.  A recent review found >1000 papers consider this (Keane & Kenny ICCBR-2019), so it’s a significant option to be improved.  Indeed, it may be the XAI strategy with the most user support, and many recent top XAI-ML publications (e.g., by Been Kim & Cynthia Rudin) use this strategy.  So, we have no regrets for CCRs *only* applying to it. The motivation for the current work is to improve the information given by the example by highlighting regions that are feature-relevant to the classification.  Though CCRs are inspired by the visual-explanation literature (e.g., saliency maps), they are a wholly new departure in **post-hoc** example-based explanation (n.b., Chen et al.~2018 NeurIPS did this **pre-hoc**), and may have potential to open a new research area sub-field.  Perhaps, we should also emphasize the widespread applicability of SP-CCRs to a wide variety of black-box classifiers, as they are ANN agnostic, but we test them here on CNNs so we can correctly compare all methods.
>
> **Revision:** Made this clearer by re-writing Section 1/2 and adding (para. 4).
> ***
>
> ### Concern #2. Position with Prototypes
> Agreed.  We did reference and point to the similarity between Chen et al. and CCRs at two points in the paper. However, we can expand upon it and reference this family of methods more thoroughly.
>
> **Revision:** Expanded lit. review (para. 4).
> ***
>
> ### Concern #3. Fig 2.
> We labelled this as “Class Logit” referring to the original predicted class. We relabelled it to make it clearer. If the prediction changes from the original predicted class, we don’t alter anything, as our focus is always on the class in question (i.e., a factual explanation). Note this is what other methods do also (e.g., LIME).
>
> **Revision:** Changed “Class Logit” to “Predicted Class Logit” in Fig. 2, and mentioned what happens when the prediction changes in Section 3.
> ***
>
> ### Concern #4.  Expt. 2 Doesn’t Evaluate Training Examples:
> You are correct, the evaluation is primarily linked to assessing the relevance of the regions added, we are not concerned about specific training examples in this evaluation. Our view is all training examples are used to train the CNN, so all have some connection to every prediction (albeit in vastly different proportions). To find the most important training images for a prediction, we don’t rely on this evaluation, we rely on a good explanation-by-example method (hence the need for the Appendix A experiment we did).
> ***
>
> ### Concern #5. Are features really “unlearned”?
> We had considered this, and recently ran experiments to help verify this. Specifically, we re-ran Expt. 2 by gradually **occluding** the important regions rather than gradually **including** them. This included fine tuning  with much more of the image occluded (i.e., letting alpha/beta approach infinity, rather than 3 in the original version), it showed the network being reduced to random guessing when all the images were occluded (verifying that the features are “unlearned”). We hope this addresses your concerns.
>
> **Revision:** We replaced Expt. 2, and made the “unlearning of features” clear in Section 4’s intro.
> ***
>
> ### Concern #6. Expt. 2 doesn’t evaluate original CNN causality
> Agreed, this is an assumption we made, but it was nonetheless based on large empirical evidence. Note that similar tests were done and widely accepted by Hooker et al. (NeurIPS 2019; A Benchmark for Interpretability Methods in Deep Neural Networks).
>
> **Revision:** We replaced the word causality with “linked”, and made this empirically-based assumption clear in the results of Section 4.
> ***
>
> ### Concern #7. Contribution Novelty Claims
> We will remove the second claim to novelty regarding the computational evaluation designs. However, when referring to the user study, the design is wholly unique, and may be a useful template for other researchers when evaluating a specific explanation strategy (e.g., we know of no other studies which counterbalanced) so we will leave that if the reviewer agrees.
>
> **Revision:** We removed the second novelty claim.
> ***
>
> ### Concern #8. Fig. 2
> Sorry, we will adjust it in the new version.
>
> **Revision:** Redid Fig. 2.

---

> > ### Comment · Reviewer_emYe · 2021-11-19
> > **re:  Author(s) Rebuttal**
> >
> > Thanks for addressing my feedback, to a good extent your response addresses my initial concerns.
> >
> > Regarding concern#4, if indeed the focus is on assessing how well a method attributes relevance to given regions and not the relevance of the training examples per se, then, it might be better to rephrase the sentence "This procedure gives us an indication of how much of the training data images are actually responsible for test predictions", present in Section 4, in order to avoid misunderstanding.
> >
> > In addition, reading at other reviews, I share the thoughts from Reviewer jgiA regarding the performance of the proposed method on datasets that contain multiple instances of the class of interest. At this point it is no possible to make a conclusive argument about this since the used dataset (CUB and ImageNet) are to a large extent single-instance datasets. At this point, this is a limitation of the presented analysis and something that could be further explored.

---

> > > ### Author Response · Authors · 2021-11-22
> > > **Author(s) Second Response**
> > >
> > > We thank the reviewer for their engagement with us and acknowledging we addressed most of their concerns, we clarify your additional two worries below:
> > >
> > > ***
> > > *Regarding concern#4, if indeed the focus is on assessing how well a method attributes relevance to given regions and not the relevance of the training examples per se, then, it might be better to rephrase the sentence "This procedure gives us an indication of how much of the training data images are actually responsible for test predictions", present in Section 4, in order to avoid misunderstanding.*
> > >
> > > Thank you for this suggestion.
> > >
> > > **Revision:** We have taken your advice and rephrased this sentence in the final updated draft, we have also reiterated the purpose of the Appendix A experiment here for complete clarity in the same sentence also.
> > > ***
> > >
> > > *In addition, reading at other reviews, I share the thoughts from Reviewer jgiA regarding the performance of the proposed method on datasets that contain multiple instances of the class of interest. At this point it is no possible to make a conclusive argument about this since the used dataset (CUB and ImageNet) are to a large extent single-instance datasets. At this point, this is a limitation of the presented analysis and something that could be further explored.*
> > >
> > >
> > > Although our SP-CCR method is slightly different to LIME, the fundamental idea of perturbing the image with reference to a specific output logit is the same, and LIME has ample evidence it works on datasets such as COCO [2]. In addition, CAM has also been demonstrated to work on such datasets many times [1]. Hence there is no technical reason why locating test image CCRs (i.e., SP-CCRs and CAM-CCRs) should have an issue here. For example, if there are two dogs in an image labelled as “Dog”, CCRs will simply highlight the part of highest saliency as usual (either with “boxes” or superpixels).
> > >
> > > For the second part of our algorithm, when locating the NN-CCR, again any explanation-by-example method can easily find a pool of candidate nearest neighbors (so this is a non-issue), and they can be “scanned” as usual to locate the closest match to the test image CCR as described in Section 2. Although we haven’t tested such datasets explicitly, for what it’s worth, we can think of absolutely no technical reasons why CCRs shouldn’t work, as the method is so general.
> > >
> > > Where it is perhaps more interesting is if there are multiple *different* instances in an image (i.e., multi-label classification). For an image with “Dog” and “Cat” in it, for SP-CCRs you simply run the algorithm as usual, but ignore all other logits other than the one you’re interested in (dog or cat here), essentially treating it like a multiclass classification problem. With CAM you would simply attribute the activation map to the output logit of interest.
> > >
> > > As an aside, you can use LIME instead of SP-CCRs in our algorithm if this is truly a great concern (e.g., see updated Expt. 1), as it’s shown to work on COCO [2]. However, we developed SP-CCRs partly for speed (searching e.g., 50 nearest neighbors for a NN-CCR with LIME takes a long time), and so evaluation of LIME on ImageNet in Expt. 2 was not possible (n.b., it could take several months to a year to run depending on hardware and hyperparameter choices). But note that for a single explanation, it would only take a few minutes to get a CCR explanation with LIME if desired instead of using SP-CCRs when searching 10-50 nearest neighbors.
> > >
> > > As a final point, we also feel compelled to point out that most seminal XAI papers (e.g., LIME, CAM, T-CAV, ProtoPNet etc.) could also be criticised for not exploring a specific domain in their respective papers, as it is quite challenging for a short conference paper to do many domains thoroughly. If we had “spread ourselves too thin”, then the criticism may be *“There is no focus in this paper”*. However, although we are confident there is no technical issues, the reviewer is correct that investigation of CCR’s ability to explain multiple instances of the same class/object detection etc. would be an interesting use case for them in future computational and/or user evaluations, but if the reviewer doesn't mind we leave that for future work.
> > >
> > > **Revision:** Include LIME as an option in our Python library as a slower alternative to SP-CCRs if desired by the user.
> > > ***
> > > [1] Vasu, B., Rahman, F.U. and Savakis, A., 2018, June. Aerial-cam: Salient structures and textures in network class activation maps of aerial imagery. In 2018 IEEE 13th Image, Video, and Multidimensional Signal Processing Workshop (IVMSP) (pp. 1-5). IEEE.
> > >
> > > [2] Sejr, J.H., Schneider-Kamp, P. and Ayoub, N., 2021. Surrogate Object Detection Explainer (SODEx) with YOLOv4 and LIME. Machine Learning and Knowledge Extraction, 3(3), pp.662-671.

---

### Official Review · Reviewer_zNGM · 2021-11-02

**Correctness:** 3
**Technical Novelty And Significance:** 3
**Empirical Novelty And Significance:** 3
**Recommendation:** 5
**Confidence:** 3

**Main Review:**

I appreciate the wide array of experiments conducted by the author. The manuscript reflects the thought process the authors had while developing their ideas, touching multiple times at the hypotheses they had and the rational behind their algorithmic choices and experiment design.

I like the fact that the authors algorithm help explore different explanations based on different CCRs (Figure 10 and Figure 11).
Does the algorithm support finding NNs based on multiple CCRs at once, so that two images can have a high match if they share multiple CCRs? If this is the case, I strongly encourage the authors to provide examples, because such ability helps make their method more generic: Many classification decisions are based on multiple cues in the input image not a single CCR.

I would be helpful if the authors can describe how their solution can be used in practice. In particular, do the users have to pick alpha and beta? The authors provide some analysis on ImageNet and CUB-200, highlighting the compromise between higher values and lower values, and concluding that 1.25 is the optimal value for ImageNet while 1.1 is the optimal value for CUB-200. I do not fully understand how the authors came to these values and whether the users of their algorithms will be able to make informed decisions about these values (especially also due to the computational overhead of scanning a range of values).

I miss a comparison with influence functions (Koh and Liang, ICML'17). The authors mentioned that such comparison would be very computationally expensive, however, there are two algorithms that allow computing influential instances more efficiently, described in:
- Estimating Training Data Influence by Tracing Gradient Descent (Garima et al., NeurIPS'20)
- Representer Point Selection for Explaining Deep Neural Networks (Yeh et al, NeurIPS'18)
I also miss a comparison with counter-factual methods (e.g. Goyal et al., ICML'19).

Language / formatting issues:
- hyperparamter
- were another => where
- the only method that (1) use […] and (2) do => (1) uses […] and (2) does not
- misclassifcation
- Section 3 determines […] => end with a full stop.
- was tested on CIFAR-10/ImageNet => on CIFAR-10 and on ImageNet
- is the regions found are contingent => is that
- it’s effect
- [similar to Hooker et al. (2019)] => avoid text within citations.
- [Kenny and Keane 2021]  add page number instead of tbd.

Wording Suggestions:
- Critical Classification Regions => Classification-Critical Regions?
- seriously explored / considered => directly? exntesively studied? [seriously might imply unserious efforts].
- Twin-Systems, Twin Systems, twin-systems, twin-system => use a consistent term, e.g. Twin-Systems Framework (what about an abbreviation?).
- we found four candidate => ... we identified four candidate ...

**Summary Of The Paper:**

The authors proposes an approach to improve explanation-by-example techniques, by picking nearest neighbors (NNs)  based on fine-grained image content that are critical for the classification decisions. Four variations to select these NNs are explored and compared against each other.
The main advantage claimed is the causal role of the generated explanation. The authors support this claim via quantitative evaluation and human evaluation.


**Summary Of The Review:**

The manuscript is heavy on analysis and seems to reflect work-in-progress results rather than solid and generalizable findings.

---

> ### Author Response · Authors · 2021-11-18
> **Author(s) Rebuttal**
>
> We thank the reviewer for the very insightful review, we address all your concerns below and how the manuscript was revised:
> ***
>
> ### Comment#1.  Does the algorithm support finding NNs based on multiple CCRs at once?
> Thank you for noting these images (Fig. 10/11) in the Appendix, we also initially felt that showing multiple CCRs would be better to tell a more comprehensive story (so we tried to at least demonstrate it in these images for when different NN-CCRs differ by small L2 distances). Crucially however, the reason the paper focused on just one CCR was because we didn’t want to overburden people in the user study, and the computational evaluation mimicked this to be consistent overall. Hence, the algorithm unfortunately doesn’t currently support what you suggest.  However, it is a viable option to explore and possibly improve CCRs in the future.
>
> **Revision:** Mentioned in the conclusion as future work.
> ***
>
> ### Comment#2.   How to use CCRs and pick hyperparameters in practice?
> Good point, we examined this for the revision.  Expt. 2 was originally designed to be consistent with Expt. 1 when parts of the image were gradually **included**. However in recent weeks we have explored the gradual **occlusion** of the important regions instead (as an aside we also did this in Expt. 1 for the revision), and have found the results far more illuminating in Expt. 2. It seems that even if one important region is included in the model during fine-tuning (even in the random variant), it is sufficient for the accuracy to be largely maintained, but by occluding them all together, the model never has a chance to see the feature again, it “unlearns it”, and the results are more discriminatory.  These new results show that the beta parameter should tend to infinity (i.e., defined here as just considering all positive superpixel regions as potential NN-CCRs), whilst alpha can be any value > 1 (but we recommend 5 to be consistent with Zhou et al’s original CAM paper, as it isolates an important object well and will inevitably tend to focus on more discriminatory regions). As such, we hope the reviewer agrees that these re-writes constitute a significant and serious response to this concern.
>
> **Revision:** We have replaced Expt. 2 (and Expt. 1) with these new results as well as including a new “Computational Conclusions” paragraph heading with definitive conclusions on generalisable hyperparameters.
> ***
>
> ### Comment#3.  Why no comparison with influence functions or Goyal counterfactual?
> We omitted influence functions from the Appendix A experiment because, apart from their high compute overhead, our preliminary tests showed they often retrieve examples which are classified to be in a different class to the test image. For instance, the nearest neighbor explanation may say *“I predicted this image as a bird because it looks like this training image I learned to predict as a truck”*.   Interestingly, twin-systems avoided this problem (see Appendix A experiment “Agreement” metric, and prior work on twinning).  Accordingly, influence functions may be unsuitable for explanations. However, if the reviewer feels very strongly, and pending acceptance, we commit to including them in the Appendix A experiment for the CRC (using one of the faster variants you suggested).
>
> **Revision:** We await your advice.
>
> Regarding the Goyal et al. counterfactual method, it is a very different type of explanation (however it is true that they do share similarities with ours, such as using “parts” of training images). The issue is that the algorithmic objective to find a replacement part of an image to modify the classification (i.e., a Goyal counterfactual) is very different to one which provides a factual build-up of evidence as to why the classification was made in the first place (i.e., our CCR method), hence the regions found will inevitably be quite different between methods. We chose to focus on factual explanations, not counterfactual explanations, esp. as the latter are quite different from a user perspective. Computationally, this may also be impossible, as Goyal's method hasn't been shown to scale to ImageNet. Lastly, whilst it is theoretically possible to compare Goyal’s method to the latent-based CCRs (as they both focus on small “box regions”), it is unclear to us how we would fairly compare Goyal’s method to the superpixel one here. We will make this clearer in the introduction and cite the relevant papers you mention.
>
> **Revision:** We mention the difference to Goyal et al. in the literature review.
> ***
>
> Comments #4 and #5.  Language / formatting issues and Wording Suggestions:
>
> **Revision:** Thank you for the language and wording observations, all are corrected in the revised version of the manuscript.

---

### Official Review · Reviewer_jgiA · 2021-11-05

**Correctness:** 2
**Technical Novelty And Significance:** 2
**Empirical Novelty And Significance:** 1
**Recommendation:** 3
**Confidence:** 4

**Main Review:**

Strengths:

The main reason to accept this paper is empirical results, showing performance on the variants of the proposed method.  The author has evaluated performance on a standard dataset. The problem statement is exciting but not the solution.

Weaknesses:

1. The author has proposed the critical classification regions (CCR) method, but it is not discussed in the paper. How does the proposed method help to find critical regions? Does the region help in explanation?

2. The core idea is not Novel. Nearest-neighbor classification[1,2] already appears as an explanation for visual object recognition. Nearest-neighbor-based explanations are already presented in the literature. So the main contribution is not novel and seems to be very limited.
In the past, Patro et al. have proposed an attention-based nearest neighbor technique to highlight specific parts of the image and try to improve attention and classification score.

3. What would be the motivation for the Nearest-neighbor example for the proposed task? Can it highlight specific parts in the classification to enhance the explanation? What is its requirement? Is this the only method to solve the task? The author should explain this.

4. The core idea is not Novel. Nearest-neighbor classification[1,2] already appears as an explanation for visual object recognition. Nearest-neighbor-based explanations are already presented in the literature. So the main contribution is not novel and seems to be very limited.

5. What would be the motivation for the Nearest-neighbor example for the proposed task? Can it highlight specific parts in the classification to enhance the explanation? What is its requirement? Is this the only method to solve the task? The author should explain this.

6. Selection NN for test samples: This part is not discussed with a detailed analysis. The method section is not clearly discussed. The author should rewrite these sections.

7. How does Speeding-up Twin-Systems help improve explanation? If not, please remove that from the method section; it creates confusion.

8. What happens when the method gets a different pattern/object/class in the nearest neighbor? Is there any way to tackle this issue? Have you tried multiple nearest neighbors like K=1,2,3,4…

9. Patro et al.[3] have tried different values of K and used a clustering technique to select the optimal nearest neighbor. They also tried contrastive techniques using positive examples, negative examples to anchor to improve the critical regions.

10. The proposed method has not been compared with the latest explanation methods. The author should provide both quantitative and qualitative comparison results with the latest explanation methods and also visualize the proposed explanation method with the latest methods.

11. The author should explain figure-6 and its attributes; it is not clear from the caption. What do you mean by those plots and the significance of those bars?

12. CUB and Imagenet datasets are mainly focused on single instances. What happens to your algorithm for multiple instance data like MS-COCO images.


13. However, the paper misses one of the core aspects of machine learning practice: readability and reproducibility of results. The author should provide an algorithm or pseudocode to reproduce the results, missing in this paper.

Ref:
[1] Chen, George H., and Devavrat Shah. "Explaining the Success of Nearest Neighbor Methods in Prediction." Foundations and Trends® in Machine Learning 10, no. 5-6 (2018): 337-588.

[2] Ma, Wei, Kendall Nowocin, Niraj Marathe, and George H. Chen. "An interpretable produce price forecasting system for small and marginal farmers in India using collaborative filtering and adaptive nearest neighbors." In Proceedings of the Tenth International Conference on Information and Communication Technologies and Development, pp. 1-11. 2019.

[3] Patro, Badri, and Vinay P. Namboodiri. "Differential attention for visual question answering." In Proceedings of the IEEE conference on computer vision and pattern recognition, pp. 7680-7688. 2018.



**Summary Of The Paper:**

The author focuses on exploring the XAI methods that none of the methods discussed, highlighting specific parts in the classification to enhance the explanation. The author proposes a method, Critical Classification Regions (CCRs), to do this. CCRs use a nearest-neighbor example to highlight similar important parts in the image explanation. The author performed a user study on a subset of the Imagenet to show improvement of the CCR method.



**Summary Of The Review:**

Overall, I do not feel like this paper is above the bar for acceptance because the core idea is not terribly exciting and has a lot of technical issues.
The author should rewrite the paper and provide all other details mention in the weakness section.

---

> ### Author Response · Authors · 2021-11-18
> **Author(s) Rebuttal**
>
> We thank the reviewer for their insights, we reply to each numbered weaknesses; w#3 = Weakness numbered (3), and how we altered the paper in our revised submission.
>
> ***
> **w#1.**  We do not understand this criticism.  Section 1/2 discuss the CCR method in detail, such as how it “helps to find” critical regions.  Also, the user study shows how the region “helps in explanation”.   We ask the reviewer to please read these sections.
> ***
> **w#2/4.**  [1,2] are mostly about k-NN in general, not “CCR-like” explanations. [3] is relevant as a heat-map method in visual question-answering, but this is quite distinct from the XAI literature.  Moreover, this is very similar to the CAMs/FAMs in our tests; as we point out there are issues with these that CCRs are designed to solve  (e.g., it is hard to know where features begin and end with many highlighted regions).  Furthermore, [3] (like others) is quite specific and does not generalise to any “ANN black-box”, whereas SP-CCRs should (but we test them on CNNs to compare all methods). This is one of the novel, significant contributions made by our paper.
>
> **Revision:** We cited [3] and fine-hone the claimed novelty (para. 4).
> ***
>
> **w#3/5.**  For 20+ years, case-based reasoning has argued k-NN can naturally explain predictions using examples.  As you say, example-based-explanation is not novel. The motivation for CCRs is to improve the information given by the example by highlighting relevant parts in the classification.  So our contribution takes insights from the visual-explanation literature (i.e., saliency methods) and applies it to supplement example-based explanations.  So, CCRs better surface the decisions of a black-box classifier. For instance, CCRs expose bias (e.g., Fig. 5b and Fig. 12a/b). The competing methods to solve this are competitively tested (Section 2-3).
>
> **Revision:** We stated the historical context and contributions better (Section 1).
> ***
>
> **w#6.**  The reviewer is correct.  We have moved the section on twin-systems to Appendix B and re-written Section 2.
> ***
> **w#7.**  The reviewer is correct that speeding-up twin systems is of no consequence to explanation.  We felt the x1000 speedup was significant still (e.g., see Sokol & Flach 2020 FAccT).
>
> **Revision:** Twin-systems speedup section was removed.
> ***
>
> **w#8.**  Yes, this is a good idea and we have experimented with multiple neighbors in explanations and included examples (Fig. 10/11). We have not observed the method retrieving a different pattern/object in the nearest neighbor, provided alpha and beta are set to the values suggested, the explanation “regions” are qualitatively the same “feature”.
> ***
> **w#9.**  Agreed this is a good idea, similar in spirit to [3] we used a “pool” of nearest neighbors to search for the NN-CCR, rather than just the closest.  We observed this produced better explanations. We did not consider contrastive explanations, as the consensus is that they involve a different set of explanatory requirements, that is a future direction.
>
> **Revision:** Said this in the conclusion.
> ***
>
> **w#10.**  We have compared our method to the most relevant in the literature.   Recall, related work is not saliency-mapping, but post-hoc solutions to find relevant regions in nearest neighbours, and “link” them to a test image region.  There are actually very few prior methods that address this.  The closest example we found is [4] which we show for comparison in Fig. 1. [5] is also close; however, this is a very constrained pre-hoc method with prototypes, whilst ours is a generic post-hoc one with all training data, so it is not at all obvious how to compare (especially our superpixel method which is extremely different). Also, despite having 250+ citations and much follow-up work, [5] has not been demonstrated to work on more realistic datasets like ImageNet, so a comparison here seems impossible).
> ***
> **w#11.**  Agreed, this is unclear.
>
> **Revision:** Rewrote Appendix A experiment.
> ***
>
> **w#12.**  ImageNet has several classes in each instance, although it is labelled with one. So, our impression is that the algorithm should work for MS-COCO.
> ***
> **w#13.**  Note, we did include a “pip install” Python library to allow others to reproduce the work (and all code), so we do not fully accept this criticism.  However, the reviewer is correct in pointing out the absence of pseudo-code.
>
> **Revision:** Included pseudocode for Algorithms in Appendix E.
> ***
>
> [1] Chen, George H., and Devavrat Shah. Explaining the Success of Nearest Neighbor Methods in Prediction.
>
> [2] Ma, Wei, et al. An interpretable produce price forecasting system for small and marginal farmers in India using collaborative filtering and adaptive nearest neighbors.
>
> [3] Patro, Badri, and Vinay P. Namboodiri. Differential attention for visual question answering
>
> [4] Kenny, E.M. and Keane, M.T., 2019. Twin-systems to explain artificial neural networks using case-based reasoning
>
> [5] Chen et al., This Looks Like That

---

### Decision · Program_Chairs · 2022-01-20

**Decision:**

Reject

**Comment:**

The paper proposes a method to improve explanation-by-example by identifying important parts of the image when using nearest engihbor explanations-by-example. Towards this goal, the notion of Critical Classification Regions (CCR) is proposed. The method is tested both computationally and a user study.

The reviewers felt that the paper had interesting ideas, but overall the reviewers agreed that the paper needs more work before being ready for publication: this includes improving the soundness of the empirical evaluations and clarifying the contribution of the paper.